# Continuous Gaussian Measurements of the Free Boson CFT: An Exactly Solvable Model

Y. Minoguchi[1][⋆], P. Rabl[1] and M. Buchhold[2]

**1** Vienna Center for Quantum Science and Technology, Atominstitut, TU Wien, 1040 Vienna, Austria

**2** Institut für Theoretische Physik, Universität zu Köln, D-50937 Cologne, Germany

⋆ yuri.minoguchi@gmail.com

August 9, 2021

## Abstract

Hybrid evolution protocols, composed of unitary dynamics and repeated, continuous or projective measurements, give rise to new, intriguing quantum phenomena, including entanglement phase transitions and unconventional conformal invariance. We introduce bosonic Gaussian measurements, which consist of the continuous observation of linear boson operators, and a free Hamiltonian evolution. The Gaussian evolution is then uniquely characterized by the system's covariance matrix, which, despite the stochastic nature of the hybrid protocol, obeys a deterministic, nonlinear evolution equation. The stationary state is exact and unique, and in many cases analytically solvable. Within this framework, we then consider an elementary model for quantum criticality, the free boson conformal field theory, and investigate in which way criticality is modified under a hybrid evolution. Depending on the measurement protocol, we observe scenarios of enriched quantum criticality, characterized by a logarithmic entanglement growth with a floating prefactor, or the loss of criticality, indicated by a volume- or area law entanglement. We provide a classification of each of these scenarios in terms of real-space correlations, the relaxation behavior, and the entanglement structure. For each scenario, we discuss the impact of imperfect measurements, which reduce the purity of the wave function, and we demonstrate that the measurement-induced characteristics are preserved also for mixed states. Finally, we discuss how the correlation functions, or even the complete density operator of the system, can be reconstructed from the continuous measurement records.

# 1 Introduction

Quantum critical behavior, emerging from the competition between a set of non-commuting operators, is a paradigmatic and fascinating phenomenon in many-body quantum systems [1–4]. The competition between non-commuting, local operators imprints characteristic fluctuations, which leave their traces in observables up to the largest length scales. Yet, the long-wavelength behavior of critical systems is often well-captured by effective and non-interacting degrees of freedom [5,6]; a consequence of emergent symmetries at the critical point, including scale (or conformal) invariance [7–12].

A new type of critical quantum states has been recently discovered in hybrid evolution protocols, which are composed of unitary time evolution and repeated, local measurements [13–60]. Here, the competition arises between the unitary dynamics, which lead to scrambling of information and generate entanglement, and local measurements that extract information from the system, and therefore reduce its entanglement over time. Under the hybrid evolution, an initial pure state wave function remains pure, and undergoes a phase transition from an (sub-) extensively entangled state to a disentangled, local product state when the measurement rate exceeds a certain threshold [13,14]. When the unitary time evolution is implemented by generic random circuits, and the entanglement entropy undergoes a transition from volume- to area law scaling, the underlying phase transition has been identified with classical universality classes [13,49,51,52,54,61]. However, when the time-evolution is constrained by certain symmetries, e.g., for Gaussian circuits or a free Hamiltonian, the measurement-induced transition has been associated with a quantum phase transition [34,39–41,43,44,62]. This enables the notion of quantum critical behavior, and emergent conformal invariance, in a nonequilibrium wave function, generated by the hybrid evolution protocol [51–53,61].

One way to implement the scrambling evolution is by acting random unitary gates on the wave function. Then the hybrid setup can be viewed as a quantum channel [63,64], and the phase transition is understood as a type of error correction threshold in the quantum channel capacity [65]. In this picture, the unitary gates encode information non-locally in the wave function, thereby protecting it against the local readout provided by the measurements [63, 65–67]. When increasing the read-out/error rate, the encoding fails and the channel capacity goes to zero. The transition in the quantum channel capacity can also be observed in the purification time of an initially mixed state [63,68].

In turn, if the unitary evolution is implemented by a time-independent Hamiltonian, the origin of the phase transition in terms of non-commuting operators becomes directly apparent [35–41]. Repeatedly measuring a set of local operators $\{O_l\}$ then projects the system towards a shared eigenstate of all the operators $O_l$. The Hamiltonian, however, for $[H, O_l] \neq 0$ is not compatible with the measurements, and constantly pushes the system out of the eigenstate manifold. The wave function then displays a volume law (logarithmically growing) entanglement entropy if the evolution is dominated by a generic (integrable) Hamiltonian, and an area law if instead the measurement-induced collapse into eigenstates dominates [35–41]. While this competition between non-commuting operators is reminiscent of a quantum phase transition in the ground state of a Hamiltonian, it also reveals a crucial difference; when measuring $L$ different operators $O_l$, and each operator has a number of $n_O$ different eigenstates, then, due to the probabilistic nature of the measurement process, the steady state will experience a macroscopic degeneracy of $O(n_O^L)$ compatible wave functions. This gives rise to an extensive configurational entropy in the ensemble of time-evolved wave function trajectories.

Due to the large configurational entropy, observables that are linear in the state $\rho = |\psi\rangle\langle\psi|$, when averaged over many trajectories with different measurement outcomes, do not reveal any information on the phase transition or the critical state. Rather they are completely featureless. In order to detect the features of a hybrid evolution protocol, one needs access to

(and has to perform the average over) observables that are non-linear in the state, such as the entanglement entropy or certain types of connected correlation functions [40, 44]. Such observables are generally harder to detect experimentally, which yields an additional complication for the experimental implementation and classification of measurement-induced dynamics, although some direct implementations have been proposed [51, 68] and recently have been realized [69].

In this work, we focus on quantum critical behavior, and the loss of quantum criticality, under continuous measurements. We therefore start from an paradigmatic model for quantum criticality, the free boson conformal field theory (CFT) [12], and examine how continuous measurements may either enrich or destroy its quantum critical behavior. In order to address this question, we introduce an elementary, and exactly solvable setup for the hybrid many-body evolution of bosons, which we term *Gaussian measurements*. It consists of a quadratic unitary evolution, imprinted by the Hamiltonian of the free boson CFT, and an extensive set of continuously measured operators $\{O_l\}$, which are linear functions of boson operators. This setup preserves the Gaussianity of an initial wave function (or density matrix), and enables an exact, measurement-noise free, and analytically solvable expression for the stationary state density matrix and the corresponding correlation functions. The latter is a central finding of this work and establishes the Gaussian measurements as an elementary model for hybrid, measurement-induced evolution in boson systems.

We then characterize the measurement-induced dynamics in terms of three essential signatures of the steady state: (i) the structure of spatial correlations, (ii) the asymptotic relaxation time scale, which for perfect measurements is the purification time, and (iii) the entanglement structure. Our approach includes both perfect and imperfect measurements [48]; the latter corresponding to the case where only a fraction of the measurement results is collected and thus the system evolves into a mixed state. In order to classify the entanglement structure of mixed states, we make use of the logarithmic negativity [70–73]. The signatures (i)-(iii) are then readily generalized to the case of imperfect measurements. We show that the characteristic properties of measurement-induced pure states are continuously connected to those of mixed states for small and moderate imperfections. This demonstrates the robustness of both the measurement-induced dynamics and their signatures in the presence of non-perfect measurements.

In order to address the question of whether quantum critical states can persist in the presence of measurements, i.e. for a hybrid evolution protocol, and of what types of measurements destroy it, we then examine three elementary measurement scenarios: (a) local measurements, which preserve the conformal symmetry of the free boson CFT, and either (b) local or (c) non-local measurements, which both violate the conformal symmetry. Consistent with the symmetries, the quantum critical properties survive in scenario (a). It yields a nonequilibrium stationary state with scale invariant correlations and relaxation behavior, and a logarithmic entanglement growth. However, the stationary state appears no longer to be universal but rather is characterized by a floating prefactor of the logarithmic entanglement scaling, which has been observed also for monitored fermion systems [37, 43, 44, 74–76]. In turn, for scenarios (b) and (c), the quantum critical signatures are destroyed by the measurements, and the violation of scale invariance leads to the emergence of a measurement-induced mass in the spatial correlations and the relaxation time. Interestingly, the entanglement structure of both scenarios differs strongly from each other: in case (b) the local measurements push the system into a stationary state, which is similar to a ground state of a gapped boson Hamiltonian, and whose entanglement structure obeys an area law. In case (c), however, the stationary state entanglement obeys a volume law, which is reminiscent of excited states [77]. Depending on the structure of the measurement operators, one therefore recovers the basic scenarios that have been observed in hybrid protocols of random circuits with projective measurements.

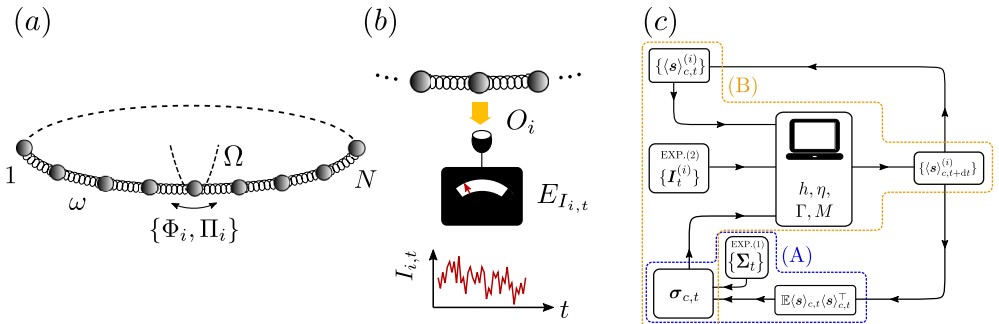

Figure 1: (a) Lattice regularization of the free boson CFT. (b) Schematic picture of measurement of operator $O_i$ located on size $i$ and the associated weak measurement operator $E_{I_{i,t}}$. The measurement outcomes are distributed following a Wiener process around the conditioned expectation value $\langle O_i \rangle_{c,t}$. (c) Sketch of the algorithm how to reconstruct the conditioned density matrix $\rho_{c,t}$. We assume that in experiment 1 (2) the quantities $\{\Sigma_t\}$ ($\{I_t^{(i)}\}$) were measured. Given the $\langle s \rangle_{c,t}^{(i)}$ for the present (initial) time the conditioned covariance matrix $\sigma_{c,t}$ is reconstructed according to Eq. (22) [see dashed box (A)]. From the present $\langle s \rangle_{c,t}^{(i)}$ and $\sigma_{c,t}$ together with $I_t^{(i)}$, we construct $\langle s \rangle_{c,t+\mathrm{d}t}^{(i)}$ by numerically applying the recursion Eq. (71) for every experimental run $i$ [see dashed box (B)].

Finally, we discuss the experimental detectability of the hybrid dynamics for Gaussian measurements. Based on the exact evolution equation, we show that a continuous recording of the weak measurement results enables a complete reconstruction of the trajectory density matrix. In a corresponding experiment it is not necessary to perform additional measurements but rather one processes the measurement outcomes that are available from generating the measurement-induced dynamics [78–81].

## 1.1 Overview

Before delving into the detailed analysis, we briefly give an overview of our work. We start with Sec. 2.1, where we provide a review of the unitary part of the evolution protocol, the *free boson CFT*. We will introduce it in the language of continuous variables [82–84], with a special focus on the *covariance matrix*, which hosts all the information on correlations in Gaussian systems, and which will be at the heart of our analysis. We then move on to discuss in Sec. 2.2 a special construction of the *stochastic Schödinger equation*, which describes the evolution of a state under both continuous unitary time evolution and continuous weak measurements. This is a non-linear stochastic equation of motion for the quantum trajectories, describing the evolution conditioned on the measurement outcomes. This framework is then straightforwardly generalized to imperfect measurements in Sec. 2.3. Equipped with this preliminaries we then introduce the notion of *Gaussian measurements*, which are bilinears in the bosonic field operators, so the hybrid evolution remains in the manifold of Gaussian states. The resulting equations of motion are found in Sec. 3.1, and they are analogous to equations well known in classical control and estimation theory. Therefore, we will refer to them as the *free boson Kalman filter*. As a key consequence, in this framework the equation of motion for the covariance matrix, a priori non-linear and stochastic, becomes non-linear but deterministic, and is described by a matrix *Riccati differential equation*. This implies that properties which inherit from the covariance matrix like correlations and the entanglement (see Sec. 3.2) are independent of the measurement-noise, and their values for an individual trajectory are identical to the trajectory average. Furthermore the steady state is obtained from solving an

algebraic equation. Given the steady state, we show in Sec. 3.3, that the asymptotic (purification) dynamics are found from linearizing the Riccati equation, giving rise to an effective non-hermitian Hamiltonian governing these.

We go on by constructing a first exactly solvable example of the measurement-induced dynamics in Sec. 4.1, where the continuously measured operators are an extensive set of linear combinations of the bosonic field operators. In Sec. 4.2 we focus on the simple limit where the fields at every site are measured and show that this scenario breaks the underlying scale invariance of the free boson CFT. This gives rise to an effective *measurement-induced mass* and consequently to fast (system size independent) relaxation and purification, short range correlations and area law entanglement (see Sec. 4.2.1-4.2.3). We then turn in Sec. 5.1, to the limit where linear combinations of the canonical conjugate of the bosonic field operators are measured in extensively many sites. In the limit where measurements are given by the momentum operators at every site, the steady state remains conformally invariant but strongly differs from the ground state of the free boson CFT. We coin this phenomenon *measurement-enriched criticality* and introduce it in Sec. 5.2. The key signatures are algebraic real space correlations, slow (system size dependent) relaxation and logarithmic entanglement growth; furthermore prefactor of the logarithmic entanglement scaling is increased, in contrast to the free boson CFT upon introducing the measurements (see Sec. 5.2.1-5.2.3). In Sec. 6.1 we numerically explore a setting where the linear combinations of the field measurements are highly non-local and drawn from a Gaussian random matrix ensemble. We present evidence that the resulting correlations functions decay exponentially and relax quickly (see Sec. 6.2-6.3). In contrast to Sec. 4 we find a volume law entanglement in the steady state.

Finally, in Sec. 7 we review the experimental difficulties of observing measurement-induced phases and criticality in general. We discuss that the conventional approach of averaging over many experimental runs erases many subtleties of the conditioned dynamics. For instance several quantities, such as the entanglement entropy for a single trajectory, which are often discussed in this context, are not accessible from a such shot measurements. In Sec. 7.2 we then show how, within the framework of Gaussian measurements, the full covariance matrix, conditioned on the measurement outcomes, may be reconstructed from the measurement records only. This enables the determination of a large set of measurement-induced characteristica, including entanglement, from the measurment record. The detailed derivations and calculations are provided in the Appendices A-E.

## 2 Setup

### 2.1 Free Boson Conformal Field Theory

We consider an elementary model for quantum critical behavior in one dimension: the free boson CFT [12]. It describes the dynamics of the massless, free Klein-Gordon field $\Phi(x)$ in $(1+1)$ dimensions via the Hamiltonian

$$H = \frac{v}{2} \int_{-\frac{L}{2}}^{\frac{L}{2}} dx \left[ \Pi^2(x) + (\partial_x \Phi(x))^2 \right]. \tag{1}$$

Here, we assume periodic boundary conditions, $\Phi(x + L) = \Phi(x)$, and a group velocity $v$. The field $\Pi(x)$ is conjugate to $\Phi(x)$ and they obey the commutation relation $[\Phi(x), \Pi(y)] = i\delta(x-y)$, which implies $\Pi(x) = \frac{1}{v}\partial_t \Phi$. In the following, we work in discrete space and consider the ultraviolet (UV) completion of the theory in Eq. (1) by putting the field on a lattice with lattice spacing $a$. This maps the continuous fields onto a discrete chain of $N = L/a$ bosonic modes with operators $\Phi(x) \to \Phi_i$ and $\Pi(x) \to \Pi_i/a$, which satisfy $[\Phi_i, \Pi_j] = i\delta_{ij}$. The corresponding

Hamiltonian describes a chain of coupled harmonic oscillators

$$H = \frac{\omega}{2} \sum_{j=1}^{N} \Pi_j^2 + (\Phi_{j+1} - \Phi_j)^2 + r_N^2 \Phi_i^2 = \frac{\omega}{2} \boldsymbol{s}^\top h \boldsymbol{s}, \tag{2}$$

with the 'spring constant' $\omega = v/a$. Here, we added a small *effective* mass, $r_N = \frac{\Omega}{\omega N}$, with $\Omega \ll \omega$, which regularizes the zero-mode of the spectrum at momentum $k = 0$, enabling well-defined inverse $H^{-1}$. The regularization scales with the system size like the finite size gap $\Delta \epsilon = \epsilon_{k_1} - \epsilon_{k=0} = O(N^{-1})$.

The Hamiltonian is a quadratic form and can be expressed in terms of the $2N$-dimensional vector $\boldsymbol{s} = (\Phi_1, \ldots, \Phi_N, \Pi_1, \ldots, \Pi_N)^\top$ and the $2N \times 2N$ Hamiltonian matrix $h$

$$h = \nu_N \oplus \mathbb{1}_N, \quad (\nu_N)_{ij} = (2 + r_N^2)\delta_{i,j} - \delta_{i-1,j} - \delta_{i+1,j}, \quad (\nu_N)_{1,N} = (\nu_N)_{N,1} = -1. \tag{3}$$

The matrix $h$ is a direct sum, including the lattice Laplacian $(\nu_N)_{ij}$ with periodic boundary conditions. In vector notation, the canonical commutation relations can be expressed in a compact way by using a vector commutator (anti-commutator) defined by $[\boldsymbol{A}, \boldsymbol{B}^\top] = \boldsymbol{A}\boldsymbol{B}^\top - \boldsymbol{B}\boldsymbol{A}^\top$ ($\{\boldsymbol{A}, \boldsymbol{B}^\top\} = \boldsymbol{A}\boldsymbol{B}^\top + \boldsymbol{B}\boldsymbol{A}^\top$). This yields

$$[\boldsymbol{s}, \boldsymbol{s}^\top] = iJ, \qquad \text{with the symplectic form} \qquad J = \begin{pmatrix} 0_N & \mathbb{1}_N \\ -\mathbb{1}_N & 0_N \end{pmatrix}. \tag{4}$$

The free boson CFT is quadratic and therefore any Gaussian state remains Gaussian under time evolution. For Gaussian states, all information about the system is encoded in the linear and quadratic moments of the vector $\boldsymbol{s}$, and higher moments can be derived from Wick's theorem. The linear moments are given by the average $\langle \boldsymbol{s} \rangle$, and the quadratic moments are collected in the symmetric covariance matrix

$$\boldsymbol{\sigma} = \frac{1}{2}\langle\{\boldsymbol{s}, \boldsymbol{s}^\top\}\rangle - \langle \boldsymbol{s}\rangle\langle \boldsymbol{s}\rangle^\top = \frac{1}{2}\langle\{(\boldsymbol{s} - \langle \boldsymbol{s}\rangle), (\boldsymbol{s} - \langle \boldsymbol{s}\rangle)^\top\}\rangle = \begin{pmatrix} \sigma_{\Phi\Phi} & \sigma_{\Phi\Pi} \\ \sigma_{\Pi\Phi} & \sigma_{\Pi\Pi} \end{pmatrix}, \tag{5}$$

where $\{\boldsymbol{s}, \boldsymbol{s}^\top\}$ is the vector anti-commutator defined above. The covariance matrix $\boldsymbol{\sigma}$ contains all possible two-point correlation functions.

For a general Gaussian initial state, the time-evolution of the first and second moments is given by the equation of motion

$$\langle \dot{\boldsymbol{s}} \rangle_t = Jh\langle \boldsymbol{s} \rangle_t, \tag{6}$$

$$\dot{\boldsymbol{\sigma}}_t = Jh\boldsymbol{\sigma}_t + \boldsymbol{\sigma}_t(Jh)^\top. \tag{7}$$

For the ground state of the free boson CFT, all the linear moments vanish, $\langle \mathbf{s} \rangle = 0$. The covariance matrix for the ground state is independent of the frequency $\omega$ and has a block structure (see [85] and App. A),

$$\boldsymbol{\sigma}_{\text{GS}} = \frac{1}{2}(\nu_N^{-1/2} \oplus \nu_N^{1/2}). \tag{8}$$

## 2.2 Stochastic Schrödinger Equation for Continuous Measurements

In order to study the effect of measurements on the free boson CFT, we subject the dynamics to continuous, weak measurements of a set of mutually commuting observables $\{O_l\}$. These are represented by Hermitian operators $O_l = O_l^\dagger$, and suitable choices shall be discussed later. We model the measurement-induced evolution of the wave function in the quantum state diffusion framework [86–89]. In this framework, the operators $O_l$ are not measured directly

but coupled to an auxilliary system, a meter, which is then read out stroboscopically at time intervals $dt$.

For each time interval $dt$, the update of the wave function $d|\psi_t\rangle_c \equiv |\psi_{t+dt}\rangle_c - |\psi_t\rangle_c$ is described by the stochastic Schrödinger equation (SSE)

$$d|\psi_t\rangle_c = \left( -iH dt - \sum_l \frac{\gamma_l dt}{2} \left( O_l - \langle O_l \rangle_{c,t} \right)^2 + \sum_l \sqrt{\gamma_l}(O_l - \langle O_l \rangle_{c,t}) dW_{l,t} \right) |\psi_t\rangle_c. \quad (9)$$

Here, $\gamma_l$ are the *measurement strengths*, i.e., the rates at which the continuous readout of observable $O_l$ changes the state over time. The nonlinear and probabilistic nature of the measurement evolution appear in the quantum state diffusion via the dependence on the instantaneous quantum mechanical expectation value $\langle O_l \rangle_{c,t} = \langle \psi_t | O_l | \psi_t \rangle_c$ and the Wiener process $dW_{l,t}$. The latter is a $\delta$-correlated white noise, which has zero mean, $\mathbb{E}(dW_{l,t}) = 0$, and variance $\mathbb{E}(dW_{l,t} dW_{m,t'}) = \delta(t - t')\delta_{l,m} dt$.

The SSE combines the unitary Hamiltonian time evolution with the infinitesimal update due to continuous measurements. Each term $\sim O_l - \langle O_l \rangle_{c,t}$ pushes the state closer to an eigenstate of the measured operator $O_l$, and is annealed in an eigenstate with $O_l|\psi_t\rangle = \langle O_l \rangle_{c,t}|\psi_t\rangle_c$. If the operators $O_l$ are pairwise commuting, $[O_l, O_m] = 0$, then there exists a set of joint eigenstates. In the absence of a Hamiltonian, $H = 0$, the system continuously involves (collapses) into such an eigenstate, which is compatible with the initial conditions. If, however, $H$ is nonzero and does not commute with the measurement operators $[H, O_l] \neq 0$, the unitary evolution competes with the measurements and prevents an asymptotic collapse of the wave function.

We will now briefly motivate the form of the SSE from a rigorous formulation of weak measurements [78, 79, 81, 90]. For simplicity, we focus on single observable $O$. Without specifying the microscopic realization of the auxiliary system, i.e., the meter, we expect that after each measurement of the meter we receive an outcome in terms of a number $I_t$, which we call 'current'. The measurement setup shall be designed in such a way that the overlap between the state $|\psi_t\rangle$ and an eigenstate $O|\lambda_O\rangle = \lambda_O|\lambda_O\rangle$ of the observable is given by a Gaussian $|\langle \psi_t | \lambda_O \rangle|^2 \sim \exp(-\alpha(I_t - \frac{\gamma}{2}\lambda_O)^2)$, whose center is determined by the current $I_t$ (for some $\alpha > 0$). Formally, this is expressed by a set of positive operators $\{E_{I_t}\}$ with

$$E_{I_t} = \left( \frac{8 dt}{\pi \gamma} \right)^{\frac{1}{4}} \exp\left[ -\frac{4 dt}{\gamma} \left( I_t - \frac{\gamma}{2} O \right)^2 \right], \quad (10)$$

with the completeness relation $\int dI_t E_{I_t}^\dagger E_{I_t} = \mathbb{1}$. After measuring the outcome $I_t$, the state is [78]

$$|\psi_{t+dt}\rangle_c = \frac{e^{-iH dt} E_{I_t} |\psi_t\rangle_c}{\| E_{I_t} |\psi_t\rangle_c \|}. \quad (11)$$

The probability for measuring a specific value $I_t$ is $\text{Pr}(I_t) = {}_c\langle \psi_t | E_{I_t}^\dagger E_{I_t} | \psi_t \rangle_c$, which yields the average $\mathbb{E}(I_t) = \int dI_t I_t \text{Pr}(I_t) = \frac{\gamma}{2}\langle O \rangle_{c,t}$ and the variance $\mathbb{E}(I_t - \mathbb{E}(I_t))^2 = \frac{\gamma}{16 dt}$. Since $E_{I_t}$ is Gaussian, $I_t$ is equivalent to an Itō random process

$$I_t dt = \frac{\gamma}{2} \langle O \rangle_{c,t} dt + \frac{1}{4} \sqrt{\gamma} dW_t, \quad (12)$$

with the Wiener increment $dW_t^2 = dt$, see App. B.1. The SSE for the quantum state diffusion process is obtained from Eq. (11) by sending $dt \to 0$ and expanding $E_{I_t}$ up to order $dt$ [for details see App. B.2].

Equation (9) is the conventional form of the SSE, for the dependence on the measurement outcomes $I_t$ appears only implicitly. In order to make it explicit, one may solve Eq. (12) for the

Wiener increment $dW_t$ and then insert the solution back into Eq. (9). Reinstating the index $l$ this yields

$$d|\psi_t\rangle_c = \left[ -iHdt - \sum_l \frac{\gamma_l dt}{2} \left(O_l - \langle O_l\rangle_{c,t}\right)^2 - 4\sum_l (O_l - \langle O_l\rangle_{c,t})\left(I_{l,t} - \frac{\gamma_l}{2}\langle O_l\rangle_{c,t}\right)\right]|\psi_t\rangle_c.$$
(13)

This evolution equation has the following, appealing interpretation: Given that the Hamiltonian $H$, the measurement operators $O_l$, the rates $\gamma_l$, the current state $|\psi_t\rangle_c$ *and* the measurement results $I_{l,t}$ are known, then the evolution step $t \to t + dt$ in Eq. (13) is deterministic. By recording the stream of currents $I_{l,t}$ and by feeding it into the SSE, one can therefore track the evolution of the state [91, 92]. Precisely this identity is the idea behind the experimental detection scheme, that we discuss in Sec. 7.1.

## 2.3 Imperfect Measurements and Unconditioned Evolution

The SSE in (9) predicts that weak continuous measurements preserve the purity of the wave function. Indeed, a complete readout of the auxiliary system at each infinitesimal time step $dt$ suppresses any build-up of entanglement between the state $|\psi_t\rangle_c$ and the meter, and ensures that the state remains pure. In reality, however, the measurement process may be imperfect, for instance due to unwanted interactions with an external bath or an incomplete readout. This will foster the wave function to start entangling with the environment, and the state to become impure. Here, we revisit the description for a continuous measurement evolution with imperfect measurements. This yields a stochastic master equation (SME), which we will use in the following sections in order to discuss the impact of imperfect measurements on both the measurement-induced dynamics, including the critical properties of the system, and on potential observables.

An incomplete measurement of the operator $O$ is commonly described by coupling $O$ simultaneously to two different meters, one which is read out (i.e., subject to a perfect measurement), and another one which is not read out [78–81]. No information is collected from the latter and the state must be described by a statistical ensemble including all measurement outcomes, i.e. the statistical average over all measurement outcomes of the second meter. To perform the average, we consider the conditioned density matrix $\rho_{c,t} = |\psi_t\rangle_c\langle\psi_t|$. Its evolution equation is obtained from the Itō convention $d\rho_{c,t} = d|\psi_t\rangle_c\langle\psi_t| + |\psi_t\rangle_c d\langle\psi_t| + d|\psi_t\rangle_c d\langle\psi_t|$ in combination with the SSE in Eq. (13). Before averaging over the second meter, the master equation is [79]

$$d\rho_{c,t} = -i[H,\rho_{c,t}]dt - \frac{\gamma_1 + \gamma_2}{2}[O,[O,\rho_{c,t}]]dt + \sum_{j=1,2}\sqrt{\gamma_j}dW_{j,t}\{O - \langle O\rangle, \rho_{c,t}\}. \quad (14)$$

Here, the $\gamma_j$ and the $dW_{j,t}$ are the measurement strengths and the Wiener increments for the perfect (unread) measurement $j = 1$ ($j = 2$). The Wiener increments for the two different meters are uncorrelated.

The imperfect measurement, i.e., the average over the outcomes of the second meter, is equivalent to the average over $dW_{2,t}$, eliminating it from Eq. (14). Introducing the total measurement strength $\gamma = \gamma_1 + \gamma_2$ and the *measurement efficiency* $\eta = \gamma_1/\gamma$, this yields the partially averaged SME (generalized to a set of measurement operators $\{O_l\}$)

$$d\rho_{c,t} = -i[H,\rho_{c,t}]dt - \sum_l \frac{\gamma_l}{2}[O_l,[O_l,\rho_{c,t}]]dt + \sum_l \sqrt{\eta_l\gamma_l}\{O_l - \langle O_l\rangle, \rho_{c,t}\}dW_{l,t}. \quad (15)$$

In the limit of a perfect measurement ($\eta = 1$) this evolution leaves $\rho_{c,t}$ in a pure state, while imperfect measurements ($\eta < 1$) immediately destroy the purity of the state, thereby

acting in a similar way as a regular bath. For $\eta = 0$ one finds the purely deterministic quantum master equation of the unconditioned evolution [93]

$$\dot{\rho}_t = \mathcal{L}\rho_t \qquad \text{with} \qquad \mathcal{L}\rho_t = -i[H, \rho_t] - \sum_l \frac{\gamma_l}{2}[O_l, [O_l, \rho_t]], \tag{16}$$

where all the jump operators $O_l$ are hermitian and $\rho_t = \mathbb{E}\rho_{c,t}$. This limit is then equivalent to the coupling of the system to a Markovian dephasing bath.

## 3 Gaussian measurements

The SSE (9) and the SME (15) provide the theoretical description for a general continuous measurement scenario, including an interacting Hamiltonian and arbitrary measurement operators $O_l$. Solving for the nonlinear and stochastic measurement-induced evolution is generally demanding and in most cases analytically intractable. For this reason, exactly solvable reference systems, for which the steady state solution is known are rare [13, 51, 52].

Here we introduce an analytically solvable setup, the *Gaussian measurement* [78, 79]. It describes the evolution of a state under the quadratic Hamiltonian Eq. (2) and linear measurement operators. The central object of the theory will be the connected two-point correlation function, or covariance matrix, $\boldsymbol{\sigma}_{c,t}$. It plays a central role for this setup because (i) for Gaussian measurements it follows a closed, deterministic evolution equation, which is analytically solvable, and (ii) due to the Gaussian nature of the evolution (enabling Wick's theorem), all higher order connected correlation functions can be expressed uniquely by the covariance matrix. The Gaussian measurement represents an extension of Gaussian theories, existing for both pure and mixed states, to a continuous measurement scenario. This provides a paradigmatic model for quantum criticality in the presence of measurements, similar to Gaussian Hamiltonians (Lindblad master equations) for closed (open) systems.

### 3.1 Gaussian Evolution and Linear Measurements

First, we provide the formal solution for a general Gaussian measurement scenario, which consists of a quadratic Hamiltonian $H$ and a general set of $N_m$ linear measurement operators $O_l$. They have unique expression in terms of the bosons via hermitian (real) matrices $h$ ($M$),

$$H = \sum_{i,j=1}^{2N} h_{ij}s_i s_j \qquad \text{and} \qquad O_l = \sum_{j=1}^{2N} M_{lj}s_j. \tag{17}$$

The SSE in Eq. (9) then is at most quadratic in the operators $s_i$, and therefore preserves the Gaussian nature of a state under time evolution. The conditioned density matrix $\rho_{c,t} = |\psi_t\rangle_c \langle\psi_t|$ of a Gaussian state is also of the Gaussian form

$$\rho_{c,t} = \rho_{c,t}[\langle \boldsymbol{s}\rangle_{c,t}, \boldsymbol{\sigma}_{c,t}] = \frac{1}{Z_{c,t}} \exp\left[-(\mathbf{s} - \langle\mathbf{s}\rangle_{c,t})^\top \mathcal{H}_{c,t}(\mathbf{s} - \langle\mathbf{s}\rangle_{c,t})\right]. \tag{18}$$

Here, $\mathcal{H}_{c,t}$ denotes the quadratic, *modular Hamiltonian*, which is connected to the conditioned covariance matrix, via the identity $\boldsymbol{\sigma}_{c,t} = \frac{1}{2}\coth(iJ\mathcal{H}_{c,t}/2)iJ$ [94]. According to Eq. (5), the covariance matrix is $\boldsymbol{\sigma}_{c,t} = \frac{1}{2}\text{Tr}(\{\boldsymbol{s} - \langle\boldsymbol{s}\rangle_{c,t}, (\boldsymbol{s} - \langle\boldsymbol{s}\rangle_{c,t})^\top\}\rho_{c,t})$, with the first moments $\langle\boldsymbol{s}\rangle_{c,t} = \text{Tr}(\boldsymbol{s}\rho_{c,t})$. The partition function $Z_{c,t} = \sqrt{\text{Det}(\boldsymbol{\sigma}_{c,t} + iJ/2)}$ guarantees that $\text{Tr}(\rho_{c,t}) = 1$. While the density matrix in Eq. (18) looks appealing, one should recall that it is conditioned on a single trajectory $|\psi_t\rangle_c$ and that $Z_{c,t}$, $\langle\boldsymbol{s}\rangle_{c,t}$ and $\mathcal{H}_{c,t}$ are time-dependent, stochastic quantities.

Instead of computing $\rho_{c,t}$, a more suitable approach for measurement-induced dynamics is to focus directly on the connected correlation functions $\boldsymbol{\sigma}_{c,t}$ from Eq. (5) [40, 62, 95]. As a key feature of Gaussian measurements, the evolution of $\boldsymbol{\sigma}_{c,t}$ becomes independent of the measurement outcomes $I_t$, i.e., it is noise-free, and therefore does not depend on individual trajectories. The time-evolution of $\langle \boldsymbol{s} \rangle_{c,t}$ and $\boldsymbol{\sigma}_{c,t}$ is obtained by applying the SME in Eq. (15) (see App. C.1), yielding the equations, which we will refer to as the *free boson Kalman filter*,

$$d\langle \boldsymbol{s} \rangle_{c,t} = Jh\langle \boldsymbol{s} \rangle_{c,t}dt + 2\boldsymbol{\sigma}_{c,t}M^\top\sqrt{\eta\Gamma}d\boldsymbol{W}_t, \tag{19}$$

$$\dot{\boldsymbol{\sigma}}_{c,t} = Jh\boldsymbol{\sigma}_{c,t} + \boldsymbol{\sigma}_{c,t}(Jh)^\top + JM^\top\Gamma MJ^\top - 4\boldsymbol{\sigma}_{c,t}M^\top\eta\Gamma M\boldsymbol{\sigma}_{c,t}. \tag{20}$$

Here $d\boldsymbol{W}_t = (dW_{1,t},\ldots,dW_{N_m,t})^\top$ is the vector of Wiener increments obeying the Itō rule $d\boldsymbol{W}_t d\boldsymbol{W}_t^\top = \mathbb{1}_{N_m}dt$. The measurement strength and efficiency is encoded in the diagonal matrices $\Gamma = \mathrm{Diag}(\gamma_1,\ldots,\gamma_{N_m}) > 0$ and $\eta = \mathrm{Diag}(\eta_1,\ldots,\eta_{N_m}) > 0$, which we allow to be different for each measured operator $O_l$, $l = 1,...,N_m$. Within this general framework, the measurement outcomes, as in Eq. (12), follow a Wiener random process

$$\boldsymbol{I}_t = \frac{1}{2}\Gamma M\langle \boldsymbol{s} \rangle_{c,t} + \frac{1}{4}\sqrt{\frac{\Gamma}{\eta}}\xi_t, \tag{21}$$

with the vector of outcomes $\boldsymbol{I}_t = (I_{1,t}\ldots,I_{N_m,t})^\top$, $M\langle \boldsymbol{s} \rangle_{c,t} = \langle \boldsymbol{O} \rangle_{c,t}$ and $\xi_t = \frac{d\boldsymbol{W}_t}{dt}$.

The evolution of the first moments in Eq. (19) is stochastic, but the second moments in Eq. (20) do not depend on $d\boldsymbol{W}_t$ and thus are deterministic. We emphasize that this result is obtained without performing a stochastic average. Rather the stochastic terms in the evolution equation of $\boldsymbol{\sigma}_{c,t}$, all appear as cubic powers of boson operators and their quantum mechanical averages add up to zero in a Gaussian state (due to Wick's theorem, see App. C.1). This yields a nonlinear but closed evolution equation, which is of the Riccati form [96]. It has in general a unique and well-defined steady state [97]. (In App. C.2 an alternative argument for the deterministic evolution is discussed.)

The deterministic evolution of $\boldsymbol{\sigma}_{c,t}$ has very important consequences: as long as one remains in the Gaussian manifold, Wick's theorem can be applied to express any correlation function, i.e., containing arbitrary powers in the boson fields, in terms of $\langle \boldsymbol{s} \rangle_{c,t}$ and $\boldsymbol{\sigma}_{c,t}$. However, since Eq. (20) predicts that $\boldsymbol{\sigma}_{c,t}$ is deterministic, the only stochastic elements entering higher order correlations are terms $\sim \langle \boldsymbol{s} \rangle_{c,t}$. For connected correlators, these terms, however, appear at most to linear order. Therefore, any steady state correlation function can be expressed in terms of $\boldsymbol{\sigma}_{c,t}$ and, at most, a linear trajectory average of $\sim \langle \boldsymbol{s} \rangle_{c,t}$. One consequence, which will be crucial later on, is that for any $n \in \mathbb{N}$,

$$\mathbb{E}(\boldsymbol{\sigma}_{c,t}^n) = \boldsymbol{\sigma}_{c,t}^n. \tag{22}$$

In case of imperfect measurements the non-linear part of Eq. (20) is suppressed with decreasing measurement efficiency $\eta < 1$. In the extreme limit $\eta = 0$, the non-linearity disappears, which yields a linear equation of motion for the covariance matrix. This is known as the *Lyupanov equation*. It is equivalent to the covariance matrix for the unconditioned state, $\rho_t = \mathbb{E}\rho_{c,t}$, which obeys Eq. (16).

The equation of motion Eq. (20) has a unique steady state $\boldsymbol{\sigma} \equiv \boldsymbol{\sigma}_{c,t\to\infty}$, which is independent of the initial conditions $\boldsymbol{\sigma}_{c,t=0}$ and, importantly, is the same for each trajectory. We therefore drop the index for the conditioned evolution in the following. The steady state obeys time-translational invariance, $\dot{\boldsymbol{\sigma}} = 0$, such that Eq. (20) yields

$$0 = Jh\boldsymbol{\sigma} + \boldsymbol{\sigma}(Jh)^\top + JM^\top\Gamma MJ^\top - 4\boldsymbol{\sigma}M^\top\eta\Gamma M\boldsymbol{\sigma}. \tag{23}$$

Before we discuss the steady state solution of Eq. (20) for particular cases, we add that a similar set of equations emerges in the Kalman filter from classical control and estimation

theory [97–100]. This connection is discussed in App. C.3-C.4. The Kalman filter has also been applied to measurement based feedback control in quantum optics for single particle systems [79,81,91,92,101,102]. Here we generalized the Riccati equation to Gaussian measurements in a many-body framework, which enables an analytical solution for the measured free boson CFT.

## 3.2 Entanglement and Logarithmic Negativity

A convenient tool for the characterization of measurement-induced dynamics and a quantifier for how close the state $\rho_{c,t}$ is to a product state, is the entanglement between a (connected) subsystem $A = \{1, \ldots, n_A < N\} \subset L$ and its complement $\bar{A} = L \setminus A$, where $L$ denotes the list of all sites. In case of perfect measurements ($\eta = 1$) the steady state is pure and a convenient measure for the entanglement of the state are the $n$-th order Rényi entropies and the von Neumann entanglement entropy [14,15,61]. We want to perform an entanglement classification for both dynamics in the presence of perfect and imperfect measurements ($\eta < 1$). The latter yields a mixed steady state for which the notion of entanglement is more subtle [70,103,104]. We will focus on the *logarithmic negativity* $\mathcal{N}_A[\rho]$ [71,72], which is generally (besides known exceptions [105–107]) a good measure for entanglement in mixed states. In the limit of a pure state $\rho^2 = \rho$, the logarithmic negativity becomes the $\frac{1}{2}$-Rényi entanglement entropy $\mathcal{N}_A[\rho] = S_A^{(1/2)}[\rho] = 2 \ln \mathrm{Tr}(\rho_A^{1/2})$. The logarithmic negativity with respect to the subregion $A$ is

$$\mathcal{N}_A[\rho] = \ln \mathrm{Tr}(|\rho^{\top_{\bar{A}}}|), \quad \text{where} \quad \langle i_A, j_{\bar{A}}|\rho^{\top_{\bar{A}}}|k_A, l_{\bar{A}}\rangle = \langle i_A, l_{\bar{A}}|\rho|k_A, j_{\bar{A}}\rangle \tag{24}$$

denotes the partial transposition with respect to $A$, where we used $|i_A, j_{\bar{A}}\rangle \in \mathcal{H}_A \otimes \mathcal{H}_{\bar{A}}$. For a Gaussian density matrix $\rho[\langle s \rangle, \sigma]$, the logarithmic negativity only depends on the covariance matrix $\sigma$ [82–84].

In order to compute $\mathcal{N}_A[\rho]$ for a Gaussian state [82,83], first the partial transposition for the covariance matrix is performed,

$$\tilde{\sigma} = \mathcal{T}_{\bar{A}} \sigma \mathcal{T}_{\bar{A}} \quad \text{with} \quad \mathcal{T}_{\bar{A}} = \mathbb{1}_N \oplus (\mathbb{1}_A \oplus (-\mathbb{1}_{\bar{A}})). \tag{25}$$

Then, the partially transposed covariance matrix $\tilde{\sigma}$ is diagonalized according to the Williamson decomposition (i.e. a symplectic transformation $S$ with $S^\top J S = J$)

$$\tilde{\sigma} = SDS^\top \text{ with } D = \mathrm{Diag}(\tilde{\nu}_1, \tilde{\nu}_1, \ldots, \tilde{\nu}_N, \tilde{\nu}_N). \tag{26}$$

The logarithmic negativity then is

$$\mathcal{N}_A[\sigma] = \sum_{n=1}^{N} \ln \max \left\{ 1, \frac{1}{2\tilde{\nu}_n} \right\}. \tag{27}$$

Here, we focus on the half-chain logarithmic negativity, i.e., the subset $A = 1, \ldots, N/2$ with $N$ being the length of the full system. Commonly, one distinguishes three characteristic scenarios (neglecting a constant offset) [85,108,109]

$$\mathcal{N}_A[\sigma] = \begin{cases} a \cdot N^0 & \text{``area law''} \\ \frac{c}{2} \log(N) & \text{``critical''} \\ b \cdot N & \text{``volume law''} \end{cases} . \tag{28}$$

The 'critical' scenario typically appears for the ground state wave function of a critical (conformally invariant) theory. In this case the prefactor $c$ is the central charge, which is a universal

property of the underlying CFT, and $c = 1$ in the case of free bosons. In the measurement-induced dynamics, we will encounter the situation, where the entanglement entropy obeys critical scaling even though the system is not in the ground state.

Determining the logarithmic negativity from the covariance matrix is particularly appealing for continuous Gaussian measurements since $\sigma_{c,t}$ is a deterministic quantity. The same is true for the partially transposed covariance matrix $\tilde{\sigma}_{c,t}$ and, consequently, for the logarithmic negativity:

$$\mathbb{E}\mathcal{N}_A[\sigma] = \mathcal{N}_A[\mathbb{E}\sigma] = \mathcal{N}_A[\sigma]. \tag{29}$$

This result may a priori be surprising: For a stochastic quantity, such as $\rho_{c,t}$ (Eq. (18)), and an arbitrary function $f$, one generally finds the inequality $\mathbb{E}f(\rho_{c,t}) \leq f(\mathbb{E}\rho_{c,t})$. The logarithmic negativity of a Gaussian state is an important example, for which the inequality is tight. The reason is that the fluctuating first moments of the density matrix in Eq. (18) do not enter its computation. A similar result in Eq. (29) has been derived recently also in a Gaussian replica field theory for measurement-induced dynamics for Rényi entanglement entropies in Ref. [40]. It is a special property of Gaussian measurements and does not hold for general measurement setups [14, 51, 52, 61, 110].

### 3.3 Relaxation, Riccati Spectrum and Purification

Besides the entanglement structure, measurement-induced phases can be characterized by their relaxation towards the steady state [63, 111]. In our setup, the relaxation can be inferred from a set of conditioned observables $\langle O \rangle_{c,t}$, e.g., from the outcomes of the continuous measurements. By taking the Fourier transform, $\tilde{O}(\nu) = \int \mathrm{d}t e^{i\nu t}\langle O \rangle_{c,t}$ we obtain the corresponding power spectrum $S(\nu, O) \equiv |\tilde{O}(\nu)|^2$, which depends on the observable $O$ and the system size $N$. The power spectrum is characterized by its poles $\lambda_n \in \mathcal{R}_{N,O}$, with $\lambda_n = -\kappa_n \pm i\epsilon_n$, and $\kappa_n > 0$ due to stability. The real part $\kappa_n \geq 0$ describes the relaxation back towards the steady state, while the imaginary part leads to coherent oscillations with frequency $\epsilon_n$. Focusing on finite size systems, the poles are separated by a finite size gap and there will be no branch cuts. We consider the *spectrum* of the problem to be the union of the poles $\mathcal{R}_{\mathrm{spec},N} = \{\mathcal{R}_{N,O}\}_O$ with respect to all possible observables. For a deterministic, Hamiltonian (Lindbladian) time evolution, $\mathcal{R}_{\mathrm{spec},N}$ then coincides with the spectrum of the Hamiltonian (Lindbladian). Due to the additional nonlinear dependence of the measurements on the state $\rho_{c,t}$, the measurement-induced spectrum may in general also be state-dependent.

For the free boson Kalman filter, the relaxation is described by the Riccati equation (20), which is generally nonlinear in the covariance matrix. However, since the steady state of the Riccati equation is unique, the asymptotic relaxation $t \to \infty$ becomes largely independent of the initial state. In particular, if one considers small perturbations $\delta\sigma_{c,t} = \sigma_{c,t} - \sigma$ around the steady state, their relaxation is well-described by the linearized equation. For small $\|\delta\sigma_{c,t}\| \ll \|\sigma\|$, we then expand Eq. (20) up to first order in $\delta\sigma_{c,t}$. This yields

$$\delta\dot{\sigma}_{c,t} \simeq Jh_{\mathrm{eff}}\delta\sigma_{c,t} + \delta\sigma_{c,t}(Jh_{\mathrm{eff}})^\top, \quad \text{with } h_{\mathrm{eff}} = h + 4J\sigma M^\top \eta \Gamma M. \tag{30}$$

The structure is reminiscent of the Hamiltonian evolution of Eq. (7) but now with an effective, i.e. non-hermitian, Hamiltonian $h_{\mathrm{eff}}^\top \neq h_{\mathrm{eff}}$. The eigenvalues $\lambda_n \in \mathcal{R}_{\mathrm{Ricc},N} = \mathrm{spec}(Jh_{\mathrm{eff}})$ are complex $\lambda_n = -\kappa_n \pm i\epsilon_n$ and we will refer to it as the *Riccati spectrum*. At sufficiently late times, when the covariance matrix is close to the steady state, the Riccati spectrum equals the previously defined operator spectrum $\mathcal{R}_{\mathrm{Ricc},N} = \mathcal{R}_{\mathrm{spec},N}$ (each operator can be expressed in terms of the covariance matrix). The asymptotic relaxation timescale $\tau_{\mathrm{relax}}^{-1} = \min\{\kappa_n : n\}$ is then determined by the inverse of the smallest relaxation rate. For perfect measurements $\eta = 1$ the relaxation time will reduce to the purification time $\tau_{\mathrm{relax}} \to \tau_{\mathrm{pure}}$ [63, 68].

# 4 Measurement-Induced Loss of Criticality

The free boson CFT described by the Hamiltonian in Eq. (2) describes a critical, scale invariant system. One potential consequence of performing weak continuous measurements is the violation of scale invariance, and the associated loss of critiality, in the dynamics. The violation of scale invariance is then associated with the generation of a (mass) scale from measurements. We will now discuss one possible scenario where this occurs: The continuous observation of the fields $\Phi_i$. Before we consider an explicit scenario, we start with a general measurement, which involves the fields $\{\Phi_i\}$ but not $\{\Pi_i\}$. It is expressed by $N$ measurement operators of the form $O_i = \sum_j (m_\phi)_{ij} \Phi_j$, where $m_\phi$ is still an arbitrary $N \times N$ matrix. We will then focus below on a choice of $m_\phi$ that breaks the discrete scale invariance of the stochastic Schrödinger equation, and thereby generates an effective, measurement-induced mass.

## 4.1 $\Phi$-Measurements

The general solution for measurements of the operators $O_i$ defined above can be obtained readily from the free boson Kalman filter, Eq. (20). The measurement operators above then correspond to the matrix

$$M = \begin{pmatrix} m_\phi & 0 \end{pmatrix}. \tag{31}$$

The measurement efficiencies and rates are $\eta = \text{Diag}(\eta_1, \ldots, \eta_N)$ and $\Gamma = \text{Diag}(\gamma_1, \ldots, \gamma_N)$. In terms of the submatrices $\sigma_{\alpha\beta} = (\boldsymbol{\sigma})_{\alpha\beta}$ (with $\alpha, \beta = \Phi, \Pi$) the Riccati equation yields the steady state solution

$$(\Phi, \Phi): \quad 0 \;=\; \omega(\sigma_{\Pi\Phi} + \sigma_{\Phi\Pi}) - 4\sigma_{\Phi\Phi}\mu_m\sigma_{\Phi\Phi}, \tag{32}$$

$$(\Phi, \Pi): \quad 0 \;=\; \omega(\sigma_{\Pi\Pi} - \sigma_{\Phi\Phi}\nu_N) - 4\sigma_{\Phi\Phi}\mu_m\sigma_{\Phi\Pi}, \tag{33}$$

$$(\Pi, \Pi): \quad 0 \;=\; \gamma_m - \omega(\nu_N\sigma_{\Phi\Pi} + \sigma_{\Pi\Phi}\nu_N) - 4\sigma_{\Pi\Phi}\mu_m\sigma_{\Phi\Pi}. \tag{34}$$

Here, we defined $\gamma_m = m_\phi^\top \Gamma m_\phi$ and $\mu_m = m_\phi^\top \eta \Gamma m_\phi$. Equation (34) contains only $\sigma_{\Phi\Pi}$ (and $\sigma_{\Pi\Phi} = \sigma_{\Phi\Pi}^\top$) and can be solved (for details see App. D). The solution of $\sigma_{\Phi\Phi}$ and $\sigma_{\Pi\Pi}$ is then readily obtained from the remaining equations, which yield

$$\sigma_{\Phi\Pi} \;=\; -\frac{\omega}{4}\mu_m^{-1}\nu_N + \frac{1}{2}\mu_m^{-\frac{1}{2}}\left(\gamma_m + \frac{\omega^2}{4}\nu_N\mu_m^{-1}\nu_N\right)^{\frac{1}{2}}, \tag{35}$$

$$\sigma_{\Phi\Phi} \;=\; \frac{\sqrt{\omega}}{2}\mu_m^{-\frac{1}{2}}(\sigma_{\Phi\Pi} + \sigma_{\Pi\Phi})^{\frac{1}{2}}, \tag{36}$$

$$\sigma_{\Pi\Pi} \;=\; \left(\nu_N + \frac{4}{\omega}\sigma_{\Pi\Phi}\mu_m\right)\sigma_{\Phi\Phi}, \tag{37}$$

where the matrices $\mu_m^{-1}$ and $\mu_m^{-\frac{1}{2}}$ are the (pseudo-) inverse of $\mu_m$ and $\mu_m^{\frac{1}{2}}$.

## 4.2 Measurement-induced mass

Now we consider the specific choice $O_i = \Phi_i$ for the measurement operators. It breaks the discrete scale invariance of the stochastic Schrödinger equation (9) and for identical $\gamma_i = \gamma$ and $\eta_i = \eta_0$ it yields the matrices

$$m_\phi = \mathbb{1}_N, \qquad \eta = \eta_0\mathbb{1}_N \quad \text{and} \quad \Gamma = \gamma\mathbb{1}_N. \tag{38}$$

For this uniform distribution of measurement rates, the system still possesses discrete translational invariance, and it is convenient to express the covariance matrix in Eqs. (35-37) in

momentum space, $\sigma_{\alpha\beta}(q,k) = \delta(q+k)\sigma_{\alpha\beta}(q)$, with $q \in Q = [-\pi, \pi]$. This yields:

$$2\eta_0\gamma\sigma_{\Phi\Pi}(q) = \left(\eta_0\gamma^2 + \frac{\omega^2}{4}\sin^4(q/2)\right)^{\frac{1}{2}} - \frac{\omega}{2}\sin^2(q/2), \tag{39}$$

$$\sigma_{\Phi\Phi}(q) = \sqrt{\frac{\omega\sigma_{\Phi\Pi}(q)}{2\gamma\eta_0}}, \tag{40}$$

$$\sigma_{\Pi\Pi}(q) = \left(\sin^2(q/2) + \frac{4\eta_0\gamma}{\omega}\sigma_{\Phi\Pi}(q)\right)\sigma_{\Phi\Phi}(q). \tag{41}$$

### 4.2.1 Purity

In order to understand the impact of monitoring on the steady state $\rho_{c,t\to\infty}$, we start with an analysis of its purity. For a Gaussian state, the purity is computed from the covariance matrix via [82–84]

$$-\ln\text{Tr}(\rho_{c,t\to\infty}^2) = \ln\text{Det}(2\boldsymbol{\sigma}) = S_2[\boldsymbol{\sigma}], \tag{42}$$

where $S_2[\boldsymbol{\sigma}]$ is the second Rényi entropy. In the steady state described by Eqs. (40-41), each momentum mode contributes one eigenvalue $\frac{1}{\eta_0}$, and therefore

$$S_2[\boldsymbol{\sigma}] = -\frac{N}{2}\ln\eta_0. \tag{43}$$

For a perfect measurement setup $\eta_0 = 1$ and the state remains pure under the stochastic evolution. For $\eta_0 < 1$, however, the system evolves into a mixed state and the impurity scales extensively in the size $\sim N$. For mixed steady states, the $n \geq 1$ Rényi entropies serve no longer as good quantifiers for the measurement-induced dynamics [17] and we therefore make use of the logarithmic negativity.

### 4.2.2 Riccati Spectrum and Relaxation

The measurement-induced evolution approaches a unique steady state $\boldsymbol{\sigma}$, which is independent of the initial state. At large times, the asymptotic relaxation towards $\boldsymbol{\sigma}$ is described by the effective Hamiltonian $h_{\text{eff}}$ in Eq. (30). The first consequence of the measurement-induced violation of scale invariance is the generation of a relaxation scale $\kappa > 0$. It appears as a mass (or gap) in the spectrum of $h_{\text{eff}}$ and imposes a rapid, exponential relaxation towards the steady state.

For the steady state $\boldsymbol{\sigma}$ from Eqs. (39-41) the effective Hamiltonian does not couple different momentum sectors and can be expressed for each momentum mode $q$ individually,

$$Jh_{\text{eff}} = Jh - 4\boldsymbol{\sigma}_0 M^\top\eta\Gamma M = \begin{pmatrix} -4\sigma_{\Phi\Phi}(q)\eta_0\gamma & \omega \\ -\omega\sin^2(q/2) - 4\sigma_{\Pi\Phi}(q)\eta_0\gamma & 0 \end{pmatrix}. \tag{44}$$

It has complex eigenvalues

$$\lambda(q) = -\kappa(q) \pm i\epsilon(q) \quad \text{with} \quad \epsilon(q) = \sqrt{\kappa^2(q) + h^2(q)}, \tag{45}$$

which are composed of the relaxation rate for momentum $q$, $\kappa^2(q) = 2\eta_0\gamma\omega\sigma_{\Pi\Phi}(q)$, and the free boson dispersion $h(q) = 2\omega|\sin(q/2)|$ shown in Fig. 2(a). Here, the largest (smallest) decay rates correspond to the long (short) wavelength modes $q \to 0$ ($q \to \pm\pi$),

$$\kappa_0 = \lim_{q\to 0}\kappa(q) = \omega\sqrt{\sqrt{\eta_0\gamma/\omega}}, \tag{46}$$

$$\kappa_\pi = \lim_{q\to\pm\pi}\kappa(q) = \omega\sqrt{\sqrt{(\gamma/\omega)^2\eta_0 + 4} - 2} < \kappa_0. \tag{47}$$

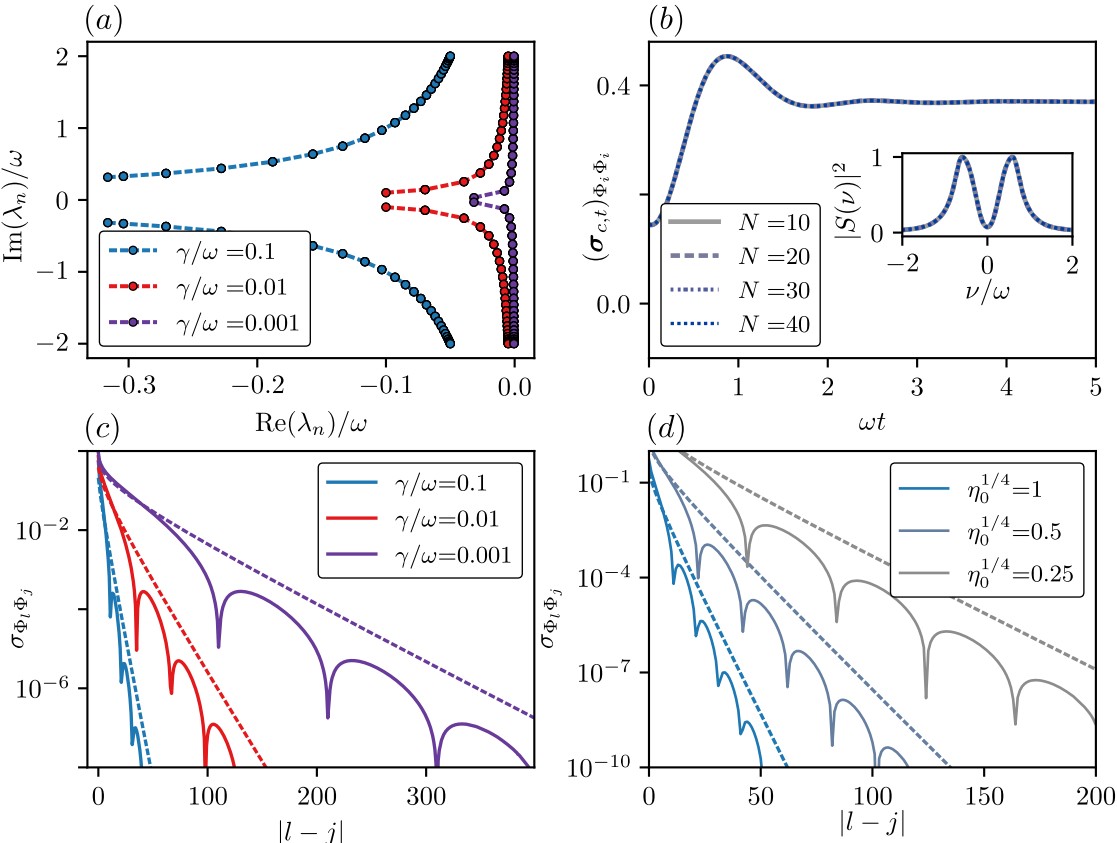

Figure 2: (a) Riccati spectrum $\mathcal{R}_{\text{Ricc},N}$ for $\Phi$-field measurements and for different measurement strengths $\gamma$. Markers correspond to results obtained from a numerical diagonalization of Eq. (30) with finite $N = 30$ and $\eta_0 = 1$; the dashed lines correspond to the analytical result from Eq. (45). The predominant feature is an emerging gap in the imaginary part of the eigenvalues corresponding to the measurement-induced mass. (b) Numerical simulation of the conditioned dynamics of the field fluctuations at $i = N/2$ after a mass quench $r = \Omega/\omega = 10$ and $\eta_0 = 0.9$ for different system sizes $N$. The inset shows the power spectrum for different system sizes $N$, where a trivial $\sim \delta(\nu)$ has been omitted. (c) Steady state real space field correlations $\sigma_{\Phi_l \Phi_j}$ for $\eta_0 = 1$ and $N = 1000$ with periodic boundary conditions for different measurement strengths $\gamma$. The solid lines correspond to exact numerical results obtained from the solution of Eq. (23). The darker lines correspond to the approximate analytical result of Eq. (48). (d) Real space correlations $\sigma_{\Phi_l \Phi_j}$ under measurement imperfections for $\gamma/\omega = 0.1$ (blue line in (c)) and different $\eta_0$. Solid (dashed) lines corresponds to exact (approximate) results.

Both rates $\kappa_\pi$ and $\kappa_0$ are independent of the system size $N$, which leads to an exponential decay of any perturbation in the thermodynamics limit $N \to \infty$. The same measurement-induced gap scale also enters the coherent oscillation frequency $\epsilon(q)$ thereby generating an effective *measurement-induced mass* $m_0 = \kappa_0/\omega$, which will have consequences for the static properties like the correlations. The slowest relaxation time scale is size independent $\tau_{\text{relax}} = \kappa_\pi^{-1} \sim N^0$. This matches the measurement-induced purification dynamics for a system in an area law phase described in Refs. [63, 68], where for perfect measurements a pure state is reached on a timescale which is independent of the system size.

The gap $\kappa(q)$ remains robust also for imperfect measurements $0 < \eta_0 < 1$. Upon decreasing the efficiency $\eta_0$ both rates $\kappa_{0,\pi}$ are reduced and the steady state is approached slower. The relaxation rate vanishes in the limit $\eta_0 \to 0$ in which a fully mixed state is approached. Imperfect measurements thus suppress the relaxation towards the steady state.

The measurement-induced relaxation scale $\kappa(q)$ can be measurement in an experiment, for instance by monitoring the relaxation of an observable $O$ towards its steady state value. In general, the relaxation scale should enter any observable but for concreteness we consider the $\Phi\Phi$-covariance, $(\sigma_{c,t})_{\Phi_i\Phi_i}$ with $i = N/2$. It can be obtained from the collected measurement outcomes $\{I_t\}$, see Sec. 7.2. We implement a setup, in which we start from the ground state $\sigma_{c,t=0} = \sigma_{\text{GS},r}$ of a gapped oscillator chain in Eq. (2), with a finite mass $r_N \to r = \Omega/\omega$. Then the mass is suddenly set to zero and a continuous monitoring according to Eq. (38) is implemented. We then track the relaxation of $(\sigma_{c,t})_{\Phi_i\Phi_i}$ over time, which is plotted in Fig. 2(b). The power spectrum is depicted in the inset of Fig. 2(b) and shows that the relaxation is system size independent. From the power spectrum the dominant poles $\lambda_n$ for this special quench can be extracted and their finite size scaling is shown in the inset of Fig. 3(d). These poles match the predictions of the Riccati spectrum $\mathcal{R}_{\text{Ricc},N}$ and demonstrate that the relaxation rate to the steady state is system size independent, as predicted by the relaxation scales $\kappa(q)$.

### 4.2.3 Correlation Functions and Entanglement

The measurement-induced mass not only sets a relaxation time scale but also a *correlation length*, which appears, e.g., in the covariance matrix. We illustrate this for the correlation function $\sigma_{\Phi_l\Phi_j}$ between the modes $\Phi_l$ and $\Phi_j$, for which the steady state is provided by Eq. (40). In order to approximate its Fourier transform into real space, we provide an analogy to the ground state of the hermitian Hamiltonian $H$ with dispersion $h(q)$: then the real space correlation function in the ground state is $(\sigma_{\text{GS}})_{\Phi_l\Phi_j} = \frac{\omega}{4\pi} \int_Q dq e^{-iq|l-j|}/h(q)$. In the presence of measurements, the covariance matrix, instead of being a ground state, then corresponds to the dark state of the effective Hamiltonian $h_{\text{eff}}$. It has a modified dispersion $h(q) \to \epsilon(q)$, displayed in Eq. (45). We therefore attempt the ansatz

$$\sigma_{\Phi_l\Phi_j} = \frac{\omega}{4\pi} \int_Q dq \frac{e^{iq|l-j|}}{\epsilon(q)} \simeq \frac{1}{2\pi} K_0(\kappa_0|l-j|/\omega) \sim \frac{e^{-\kappa_0|l-j|/\omega}}{\sqrt{\kappa_0|l-j|/\omega}}. \tag{48}$$

In the second step, we have replaced $\epsilon(q) \to \sqrt{\kappa_0^2 + \omega^2 q^2}$, as relevant for $|l-j| \gg 1$. This yields the 0-th modified Bessel function, $K_0(x)$, which decays exponentially $K_0(x) \simeq \sqrt{\frac{\pi}{2}} \frac{e^{-x}}{\sqrt{x}}$ for large $\kappa_0|l-j|/\omega \gg 1$. The solution of our ansatz in Eq. (48) is in good agreement with the true correlation function obtained from the Fourier transformation of $\sigma_{\Phi\Phi}(q)$, shown in Fig. 2(c). It confirms that the inverse correlation length $\xi^{-1} \sim \kappa_0$ is set by the relaxation rate $\kappa_0$.

For imperfect measurements, $0 < \eta_0 < 1$, we found the dependence $\kappa_0 \sim \eta_0^{1/4}$. Therefore the correlation length $\xi = \omega/\kappa_0$ increases with decreasing $\eta_0$, as shown in Fig. 2(d). This leaves us with the picture that continuous measurements of $\{\Phi_l\}$ lead to a localization of the wave function in the real space, where the degree of localization is proportional to

the measurement strength, i.e. $\xi \sim \gamma_0^{-1/2}$. Imperfect measurements, however, water down the localization and lead to a growth of the localization length with growing measurement inefficiency.

Exponentially decaying correlation functions also leave their fingerprints in the entanglement structure of the system. The logarithmic negativity $\mathcal{N}_A[\boldsymbol{\sigma}]$ grows for small subsystem sizes $A$ but saturates at length scales of the order of the correlation length. At sizes $L > \xi \sim \kappa_0^{-1}$ it then obeys an area law [108, 109]

$$\mathcal{N}_A[\boldsymbol{\sigma}] \simeq \frac{1}{2}\ln(\omega/\kappa_0), \tag{49}$$

which is shown by the blue dashed curve in Fig. 4(a) below.

In general we find that an arbitrary weak, measurement-induced breaking of scale invariance in the SSE (9) represents a relevant perturbation and generates a mass $\kappa_0$, which then dominates correlation functions and the relaxation behavior at large distances and large times. Overall, this may not be surprising for a Gaussian theory, for which introducing an additional scale usually corresponds to adding a relevant operator. Then the saturation of the entanglement entropy to an area law and an exponentially fast relaxation towards the steady state (including exponentially fast purification in the perfect measurement limit $\eta_0 = 1$ [63, 68]) is a direct consequence of the measurement-induced scale $\kappa_0$.

## 5 Measurement-Enriched Criticality

Another possible measurement scenario arises when the measurements are compatible with the scale invariance of the Hamiltonian in the SSE (9), and therefore with the conformal symmetry of the free boson CFT. In this case, we would expect, as for a ground state, that scale invariance has an impact on the correlation functions and the entanglement structure of the steady state. However, if the measurement operators $\{O_l\}$ do not commute with the Hamiltonian, the steady state will neither be the ground state nor any excited eigenstate of the Hamiltonian. Rather it corresponds to a new type of state, which gives rise to several modifications of the critical properties of the theory. We term this scenario *measurement-enriched criticality*. For perfect measurements, it corresponds to quantum criticality in a pure but non-ground state. Although there exist several possible choices for measurement operators, which preserve scale invariance, to be concrete, we consider measurements of the operators $O_l = \Pi_l$.

### 5.1 $\Pi$-Measurements

We start again by considering a general measurement involving the operators $\Pi_l$. This can be expressed in terms of a $N \times N$ matrix $m_\pi$ with

$$O_l = \sum_{j=1}^{N}(m_\pi)_{lj}\Pi_j. \tag{50}$$

The measurement strength and efficiency are collected in the diagonal matrices $\Gamma, \eta$, and again we define two matrices $\gamma_m = m_\pi^\top \Gamma m_\pi$ and $\mu_m = m_\pi^\top \eta \Gamma m_\pi$ for a compact notation. The Riccati

equation (23) then has the steady state solution

$$\sigma_{\Phi\Pi} \;=\; \frac{\omega}{4}\mu_m - \frac{1}{2}\mu_m^{\frac{1}{2}}\left(\gamma_m + \frac{\omega^2}{4}\mu_m\right)^{\frac{1}{2}}, \tag{51}$$

$$\sigma_{\Pi\Pi} \;=\; \frac{\sqrt{\omega}}{2}\mu_m^{\frac{1}{2}}(-\nu_N\sigma_{\Phi\Pi} - \sigma_{\Pi\Phi}\nu_N)^{\frac{1}{2}}, \tag{52}$$

$$\sigma_{\Phi\Phi} \;=\; \left(\mathbb{1}_N - \frac{4}{\omega}\sigma_{\Phi\Pi}\mu_m\right)\sigma_{\Pi\Pi}\nu_N^{-1}. \tag{53}$$

## 5.2  Measurement Enriched Criticality

Now we consider the aforementioned scenario of $O_i = \Pi_i$ and a uniform measurement strength and efficiency. This corresponds to

$$m_\pi = \mathbb{1}_N, \qquad \eta = \eta_0\mathbb{1}_N \quad \text{and} \qquad \Gamma = \gamma\mathbb{1}_N. \tag{54}$$

In this case, the solutions from Sec. 5.1 have a very peculiar form. They can be parametrized in terms of three real numbers $A, B, C \in \mathbb{R}$,

$$\sigma = \frac{1}{2}\begin{pmatrix} A\nu_N^{-\frac{1}{2}} & B\mathbb{1}_N \\ B\mathbb{1}_N & C\nu_N^{\frac{1}{2}} \end{pmatrix} \qquad \text{with} \qquad AC - B^2 = \eta_0^{-1}. \tag{55}$$

The exact dependence of the coefficients on the measurement parameter is

$$B = \frac{1}{2\eta_0\gamma}\left((4\gamma^2\eta_0 + \omega^2)^{\frac{1}{2}} - \omega\right), \quad C = \sqrt{\frac{\omega B}{\eta_0\gamma}} \ \text{and} \ A = \left(1 + \frac{2\eta_0\gamma}{\omega}B\right)C. \tag{56}$$

The parameter $B \sim \gamma$ describes a continuous deformation of the ground state covariance matrix of the free boson CFT [see Eq. (8)]. The latter is recovered for $B = 0$ (which implies $\eta_0 = 1$). In this case, $A = 1/C \neq 1$, which would correspond to another variant of the free boson CFT, namely the Luttinger liquid with a Luttinger parameter $K = A$ [40]. For imperfect measurements, $\eta_0 < 1$, and in the limit $\gamma \to 0$ the steady state is proportional to the ground state of the free boson CFT and we obtain $\sigma = \eta_0^{-1/2}\sigma_{\text{GS}}$. The determinant of $\sigma$ is then $\sim \eta_0^{-N}$, and we again obtain the steady state purity $S_2[\sigma] = -\frac{N}{2}\ln\eta_0$. Furthermore, due to the second equality in Eq. (55), this scaling of the determinant of $\sigma$ and the value of the purity holds for any $\gamma > 0$.

### 5.2.1  Correlation Functions

The diagonal blocks of the covariance matrix in Eq. (55) are almost identical to the correlation functions $\sigma_{\Phi\Phi}$ and $\sigma_{\Pi\Pi}$ in the ground state. They only deviate from that by a multiplication with the constants $A$ and $C$. The correlation function $\sigma_{\Phi\Phi}$ has a divergent short-distance behavior for $N \to \infty$ and we therefore consider the relative covariance matrix,

$$\sigma_{\Phi_l\Phi_j} - \sigma_{\Phi_l\Phi_l} \simeq \frac{A}{2\pi}\left[\text{Ci}(|l-j|) - \gamma_{\text{EM}} - \log(|l-j|)\right] \sim -\frac{A}{2\pi}\log(|l-j|). \tag{57}$$

Here, $\text{Ci}(x) = -\int_x^\infty dt\, \cos(t)/t \approx x^{-2}$, which is subleading in the limit $x \gg 1$ and $\gamma_{\text{EM}}$ is the Euler-Mascheroni constant. The $\Pi_l - \Pi_j$-covariance matrix is

$$\sigma_{\Pi_l\Pi_j} \simeq -\frac{C}{2\pi|l-j|^2}. \tag{58}$$

A qualitative modification is found, however, for the $\Phi - \Pi$ correlations $\sigma_{\Phi_l\Pi_j} = \delta_{lj}B$. The functions are local and vanish for $|l-j| > 0$, but acquire a short-distance or ultraviolet correction. As we show below, this short-range correction will have a profound consequence on the entanglement structure at large distances.

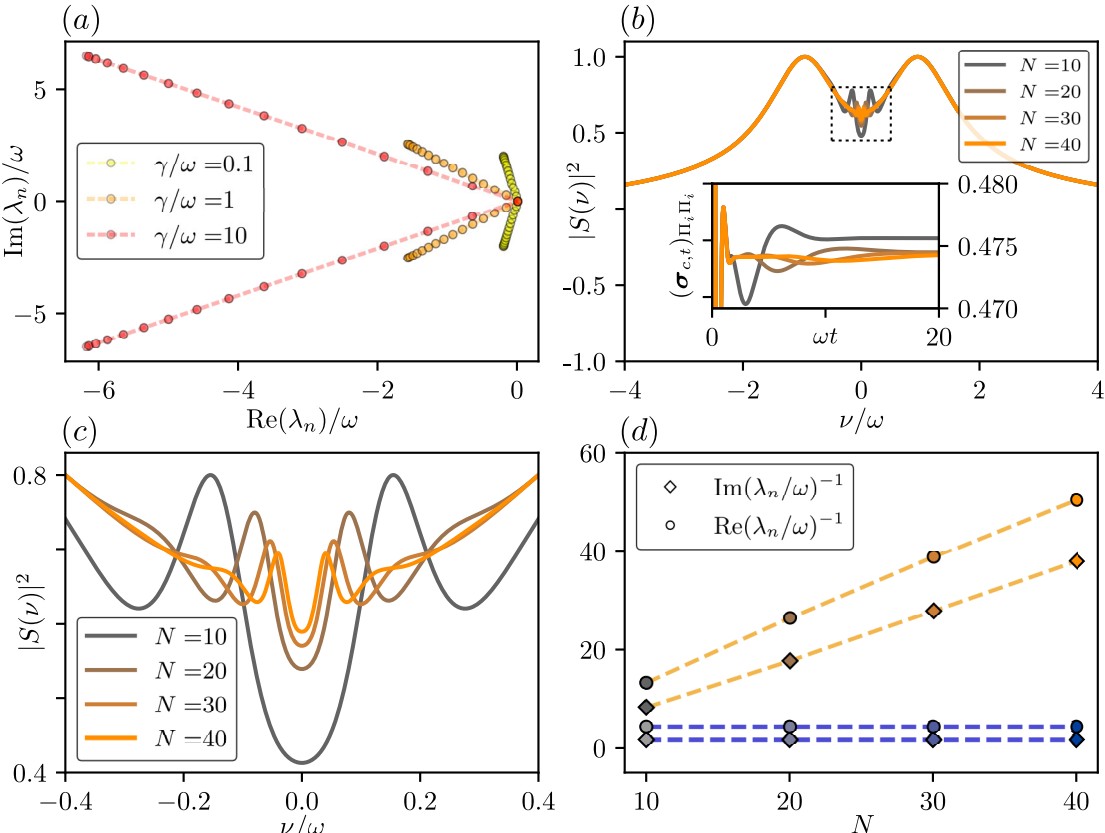

Figure 3: (a) Riccati spectrum for $\Pi$-measurements with different measurement strengths $\gamma$ at fixed $\eta_0 = 0.95$. The markers correspond to the numerical diagonalization of Eq. (59) for a system of size $N = 30$. The dashed lines corresponds to the analytical result in Eq. (60). (b) Inset: Quench dynamics under continuous $\Pi$-measurements for an initial mass quench with $r = \Omega/\omega = 10$ and $\eta_0 = 0.9$ for different system sizes $N$. Power spectrum for the quench dynamics of the inset. In (c) we zoom in on the region in the dashed box. The finite size scaling of the width (position), characteristic of the real (imaginary) part of the dominant pole is depicted. (d) Finite size scaling of the poles from the relaxation dynamics reconstructed from fitting the power spectrum of (c). The two lower blue lines depict the corresponding poles for the $\Phi$ measurements discussed in Sec. 4.2.2.

### 5.2.2 Riccati Spectrum and Relaxation

Next, we address the relaxation dynamics close to the steady state, i.e., the spectrum of the effective Hamiltonian [Eq. (30) for the steady state in Eq. (55)]

$$J h_{\text{eff}} = \begin{pmatrix} 0 & (\omega - 2B\eta_0\gamma)\mathbb{1}_N \\ -\omega v_N & -2C\eta_0\gamma v_N^{\frac{1}{2}} \end{pmatrix}. \tag{59}$$

It is diagonal in momentum space and has eigenvalues [see Fig. 3(a)]

$$\lambda(q) = -2\omega|\sin(q/2)|(h_- \pm i h_+), \qquad \text{with} \qquad h_\pm = \left(\frac{1}{2\omega}\sqrt{4\eta_0\gamma^2 + \omega^2} \pm \frac{1}{2}\right)^{\frac{1}{2}}. \tag{60}$$

For $\gamma \to 0$, $\lambda(q)$ approaches the Hamiltonian spectrum $\lim_{\gamma \to 0} h_- = 0$ and $\lim_{\gamma \to 0} h_+ = 1$. For $\gamma > 0$, the spectrum remains qualitatively similar; it is massless and linear in $q$ for small momenta. Measuring the fields $\Pi_l$, then introduces an overall constant $h_+ \geq 1$, i.e., an effective index of refraction, which increases the group velocity $\tilde{v} = h_+ v$. In the limit $\gamma \gg \omega$ it approaches $h_+ \sim \sqrt{\sqrt{\eta_0}\gamma/\omega}$, which coincides with the measurement-induced mass scale from the $\Phi$-measurement. The real part of the spectrum, describing the relaxation rate towards the steady state $\sigma$, now scales with the momentum $\kappa(q) = 2h_-\omega|\sin(q/2)| \simeq h_-\omega|q|$ for long wavelengths $|q| \ll 1$. This transfers the common dynamical scaling behavior of quantum critical phenomena, i.e., $\epsilon(q) \sim |q|$ also to the relaxation dynamics $\kappa(q) \sim |q|^z$, with a dynamical critical exponent $z = 1$.

Each momentum mode then relaxes with its own, characteristic time scale, $\tau(q) \sim (\omega h_-|q|)^{-1}$ for small momenta, i.e. large distances. In the thermodynamic limit $N \to \infty$, when the momenta $q$ are continuous, there is no upper bound for the relaxation time and the steady state is approached algebraically in time $\sim t^{-1}$. For any finite system, $N < \infty$, the largest relaxation time is set by the smallest momentum scale $q_{\min} = \pm\frac{2\pi}{N}$, $\tau(q) \leq \tau(q_{\min}) = \frac{N}{2\pi\omega h_-}$. Therefore, the overall relaxation time grows linearly with the system size. For perfect measurements, $\eta_0 = 1$, the steady state is again pure and the relaxation time is identical to the purification time, which implies a purification time scale that diverges linearly with the system size. This linear in system size scaling has also been observed at the critical point for a measurement-induced phase transition [63, 112], and we emphasize that the system for this type of measurements still purifies in the thermodynamic limit. Only this purification is not associated with a characteristic scale.

In order to discuss a potential experimental probe of the relaxation dynamics, we consider the same scenario as in Sec. 4.2.2. The system is prepared in the ground state of a gapped boson chain and then the mass gap is switched off and the measurements are switched on. Here, we instead monitor the half-chain momentum fluctuations $(\sigma_{c,t})_{\Pi_i\Pi_i}$ at site $i = N/2$, since this is what can be reconstructed from the measurement results [see Sec. 7.2]. The conditioned dynamics for different system sizes $N$ are depicted in the inset of Fig. 3(b). As we expect from our previous analysis, the early time dynamics are system size independent, since they correspond to small momentum modes. Close to the steady state, however, we observe characteristic oscillations and a decay rate, which both depend on system size. In Fig. 3(b) the power spectrum is plotted and a focus on its low-frequency part is shown in Fig. 3(c). This confirms the system size dependence of the low frequency part. However, one also observes that for larger system sizes, (i) the low-frequency peaks of the spectrum move towards $\nu = 0$ and (ii) the weight of the peaks decreases continuously. Therefore the system size dependence is continuously decreasing. We can extract the poles leading to the peaks in the spectrum by fitting a complex Lorentzian to the low frequency part of the spectrum. In Fig. 3(d) the scaling of the imaginary and real part of the dominant low frequency poles with system size is shown for $\Pi$ ($\Phi$)-measurement. It shows the characteristic scaling $\sim N$ ($\sim N^0$), matching our discussion above.

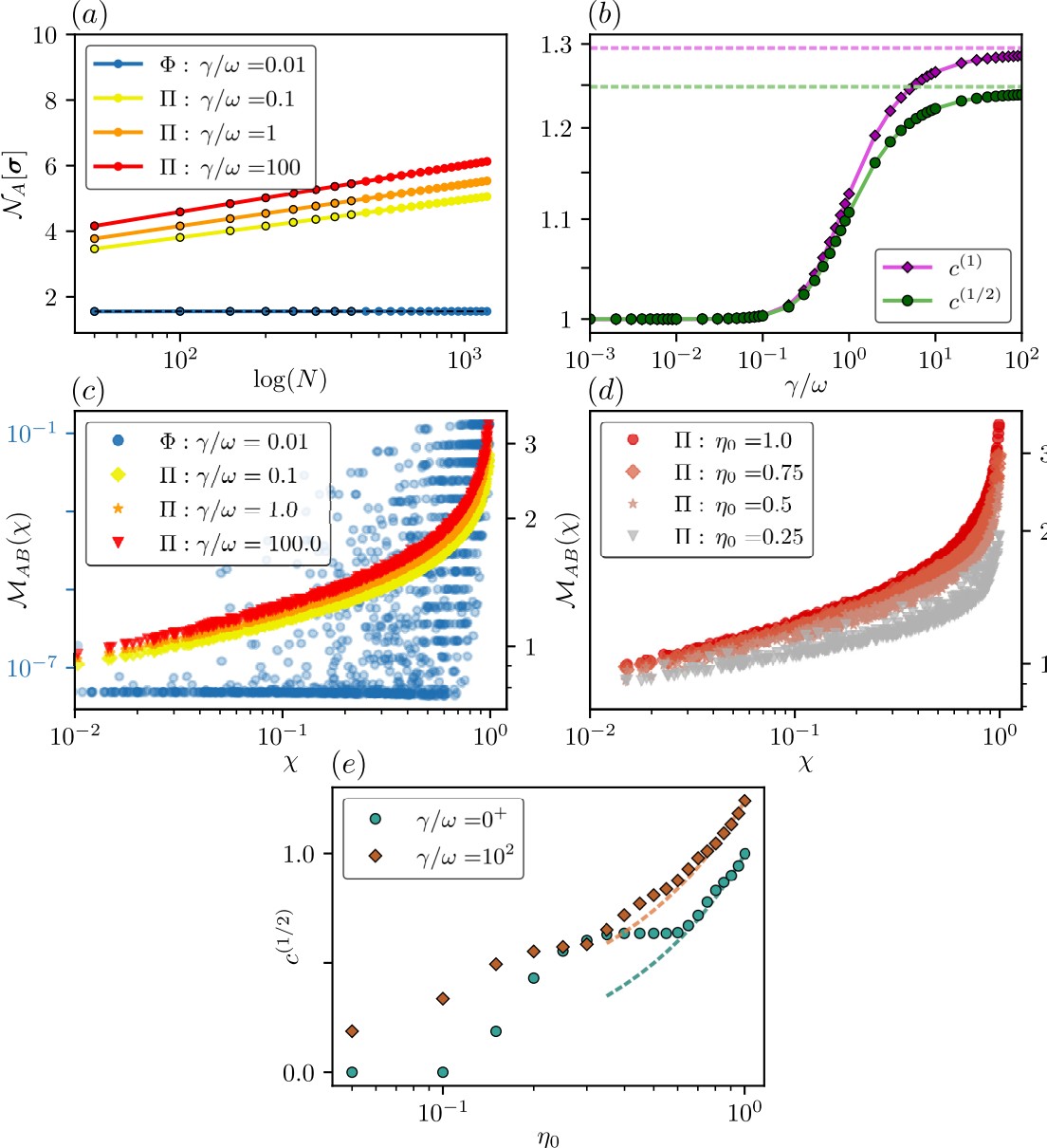

Figure 4: (a) Finite size scaling of the half system logaritmic negativity in the steady state for $\Pi$ ($\Phi$) measurements. The steady states have been obtained analytically for different system sizes $N$. For the $\Pi$ ($\Phi$)-measurements we obtain a critical (area law) scaling. The markers with dark edges correspond to the steady state obtained from the numerical solution of Eq. (23). The black dashed line (on top of the blue) corresponds to the analytical result in Eq. (49). (b) Measurement enriched criticality from sliding logarithmic prefactors $c_\gamma^{(\alpha)}$ with respect to the measurement strength $\gamma$. The purple (green) line corresponds to the prefactor $c^{(1)}$ ($c^{(1/2)}$) in the finite size scaling of the 1-Rényi entropy ($\frac{1}{2}$-Rényi entropy/logarithmic negativity). The dashed green (purple) line corresponds to the maximum attainable scaling constant $c_{\max}^{(1/2)} \approx 1.25$ ($c_{\max}^{(1)} \approx 1.29$). (c) Data collapse (non-collapse) of the mutual negativity for $\Pi$ ($\Phi$)-measurements for different $\gamma$. (d) Collapse of mutual negativity for imperfect $\Phi$-measurements at $\gamma/\omega = 100$ and different measurement efficiencies $\eta_0$. (e) Dependence of the scaling prefactor $c^{(1/2)}$ in the logarithmic negativity on the measurement efficiency $\eta_0$ for a fixed measurement strength $\gamma$. The dashed line corresponds to the linear decay of $c^{(1/2)}$ with $\eta_0$.

### 5.2.3 Entanglement and Measurement Enriched Criticality

Let us now turn to the entanglement structure of the steady state, which we characterize again by the half-chain logarithmic negativity $\mathcal{N}_A$. For this measurement scenario, we observe that $\mathcal{N}_A$ scales logarithmically with the system size [see Fig. 4(a)]

$$\mathcal{N}_A[\boldsymbol{\sigma}] = \frac{1}{2}c_\gamma^{(1/2)}\ln(N) + \text{const.} \tag{61}$$

Here, the prefactor of the log-scaling $c_\gamma^{(1/2)}$ of the logarithmic negativity ($\frac{1}{2}$-Rényi entropy), turns out to increase monotonously with $\gamma$.

We now focus on perfect measurements, $\eta_0 = 1$. Then $c_\gamma^{(1/2)}$ approaches the ground state value $c_{\gamma\to0^+}^{(1/2)} = 1$ in the limit of vanishing measurement strength and converges to an upper threshold of $c_{\max}^{(1/2)} \approx 1.25$ in the limit of infinite measurement strength. Its detailed $\gamma$-dependence is displayed by the green line in Fig. 4(b). Although the steady state for $\gamma > 0$ is not a ground state of any familiar free boson CFT, its entanglement structure is consistent with a quantum critical, i.e., conformally invariant, ground state in one dimension.

In order to test the conformal symmetry of the steady state, we rely on the characteristic form of four-point correlation functions in a conformal field theory. One possibility for such a four-point function is the mutual (logarithmic) negativity [13,17,22,53,74]

$$\mathcal{M}_{A,B}[\chi] = \mathcal{N}_A[\boldsymbol{\sigma}] + \mathcal{N}_B[\boldsymbol{\sigma}] - \mathcal{N}_{A\cup B}[\boldsymbol{\sigma}], \tag{62}$$

for the two intervals $A = [i_1, i_2]$ and $B = [i_3, i_4]$. From intervals $A, B$ one can compute the cross ratio $\chi = i_{12}i_{34}/(i_{13}i_{24})$, where $i_{ab} = \sin(\pi|i_a - i_b|/N)$. For a conformally invariant state, the mutual negativity $\mathcal{M}_{A,B} \equiv \mathcal{M}(\chi)$ depends only on the value of the cross ratio $\chi$ for any randomly chosen pair $A$ and $B$. Indeed, as shown in Fig. 4(c), for $\Pi$ measurements, the mutual negativity shows a scaling collapse as a function of $\chi$, indicating conformal invariance [13,53,95]. The collapse is absent for measurements, which induced a mass [see blue markers in Fig. 4(c)].

For ground states in (1+1)-dimensional systems, the prefactor of the entanglement entropy (the central charge) $c$ is intimately linked to the symmetries of the Hamiltonian. A sliding prefactor $c_\gamma^{(1/2)}$ (see also [62,95]) as depicted in Fig. 4(b) is peculiar. When interpreted as a central charge, it would indicate that continuous monitoring would prepare the system in the ground state of a CFT but with a larger central charge $c_\gamma^{(1/2)} > 1$. However, we believe that it is more likely that the modification of $c_\gamma^{(1/2)}$ is rooted in the fundamentally non-equilibrium character of the steady state, which may not correspond to a ground state of any established CFT. This interpretation is strengthened when considering another entanglement measure, e.g., we show the 1-Rényi entropy (or von Neumann entanglement entropy) as the purple line in Fig. 4(b). It shows that the logarithmic scaling factor $S_A^{(1)}[\boldsymbol{\sigma}] = \frac{1}{3}c_\gamma^{(1)}\ln(N)$ differs from the one of the logarithmic negativity (see [113]). If indeed the prefactor in the log-scaling of the entanglement measures was universal, i.e., as the central charge for a CFT in the ground state, we would expect $c_\gamma^{(1/2)} = c_\gamma^{(1)}$.

We will now turn to the entanglement structure for imperfect measurements, $0 < \eta_0 < 1$. In Fig. 4(d), we depict the mutual negativity under imperfect measurements, which indicates that the steady state is still conformally invariant. Upon decreasing the efficiency there is still a data collapse, however less crisp as for $\eta_0 = 1$. The logarithmic negativity $\mathcal{N}_A[\boldsymbol{\sigma}]$ for fixed measurement strength $\gamma$ but decreasing efficiency $\eta_0$ remains a logarithmic function of the system size and we can assign an effective logarithmic prefactor $c_\gamma^{(1/2)}$ even for $\eta_0 < 1$. In Fig. 4(e) the deterioration of this prefactor with $\eta_0$ is shown. We find that decreasing the measurement efficiency $\eta_0$ leads to a monotonous decrease of $c_\gamma^{(1/2)}$, counteracting the $\gamma$-induced increase.

### 5.2.4 Connection to $\Phi$-Measurements

A critical measurement-induced evolution can also be obtained by general position measurements, such as in Eq. (31). This is realized, for instance, by measuring the operators $O_i = \Phi_{i+1} - \Phi_i$ with a uniform measurement strength $\gamma$, which corresponds to

$$(m_\phi)_{ij} = \delta_{i,j} - \delta_{i-1,j} - \delta_{iN}\delta_{j1}, \qquad \eta_\phi = \eta_0 \mathbb{1}_N \quad \text{and} \quad \Gamma_\phi = \gamma \mathbb{1}_N. \qquad (63)$$

Inserting this into Eqs. (39-41) we obtain a covariance matrix as in Eq. (55), only with the constants $A$ and $C$ exchanged. Therefore, measuring $\{O_i = \Pi_i\}$ or $\{O_i = \Phi_{i+1} - \Phi_i\}$ yields similar steady states. This may not be too surprising since taking the continuum limit of the latter yields $O_i \to O(x) = \partial_x \Phi(x)$. Rewriting the Hamiltonian in Eq. (1) in terms of its dual field yields

$$H = \frac{v}{2} \int_{-\frac{L}{2}}^{\frac{L}{2}} \mathrm{d}x \left[ (\partial_x \theta(x))^2 + \Pi_\theta^2(x) \right], \qquad (64)$$

with the dual field $\theta(x) = \int_{-L/2}^{x} \mathrm{d}y\, \Pi(y)$ or equivalently $\Pi = \partial_x \theta$ [3, 114]. The momentum conjugate to the dual field $\theta$ is $\Pi_\theta = \partial_x \Phi$ and they satisfy the canonical commutation relation $[\theta(x), \Pi_\theta(y)] = i\delta(x - y)$. Since the $\theta(x) \leftrightarrow \Phi(x)$ is a symmetry of Eq. (1), measuring $O(x) = \partial_x \Phi(x) = \Pi_\theta(x)$ is equivalent to the momentum measurement discussed above.

## 6 Steady States with Extensive Entanglement Scaling

In the previous sections, we have discussed measurement protocols, which lead to steady states with either logarithmic growth of the entanglement entropy or an area law entanglement. In both cases, the entanglement structure is similar to that of a ground state wave function, either for a gapless or a gapped Hamiltonian. For generic measurement-induced dynamics, however, an extensive scaling of the entanglement entropy has been reported [13–15]. An extensive entanglement entropy, obeying a volume law growth, is typically associated with an excited state, located in the center of the spectrum [115–117]. For Gaussian measurements it is a priori not clear, whether or not states with an extensive entanglement entropy can be sustained from the interplay of local measurements and local Hamiltonians due to the structure of the reduced Hilbert space. For example, volume law entangled states for free fermions [112] as well as for free bosons [118] have been ruled out.

In order to mimic the Hilbert space structure of generic, interacting systems within the framework of Gaussian measurements, we will now turn to a measurement protocol where the measured operators $\{O_l\}$ are chosen from a random matrix ensemble. This setting is strongly idealized since the non-local measurements are experimentally challenging to implement. In this setting the measurement operators have a highly non-local structure in the eigenbasis of the Hamiltonian. The measurement-induced stationary states then correspond to highly excited states in the spectrum of the Hamiltonian and share many properties, which are reminiscent of the eigenstates of random matrices [119–121]. Those properties include short range correlations, finite relaxation times as well as extensive entanglement scaling [77].

### 6.1 Measurement Operators

For concreteness we will consider measurement operators $O_l = \sum_{j=1}^{N} m_{lj} \Phi_j$, where $m$ is a stationary but random matrix, which is drawn from the Gaussian orthogonal ensemble (GOE). It is sampled from to the distribution function $\Pr(m) \sim \exp[-\mathrm{Tr}(m^\top m)]$. For this choice of random measurement operators, the Riccati equation (23) has no particular symmetries, which

would allow us to determine the steady state solution analytically. We thus solve it for the steady state numerically.

## 6.2 Correlation Functions

In order to discuss the real space correlations in the steady state, we compute the stationary state covariance matrix and average it over several realizations of random measurement matrices $m$. The averaged correlation function $\overline{\sigma_{\Phi_i \Phi_l}}$ for $|i - l| \gg 1$ decays exponentially in the distance, featuring a correlation length $\xi$, which is shown in Fig. 5(a). This is also characteristic for excited states of many body systems which are well captured by eigenstates of random matrices [119–121].

The correlation length $\xi$ displays a power-law-dependence on both the measurement strength $\gamma$ and the measurement imperfection $\eta_0$, which is shown in Fig. 5(b). For perfect measurements, $\eta_0 = 1$, one finds that the correlation length decays with the measurement strength $\xi \sim \gamma^{-\alpha}$ with an exponent $\alpha = 0.478$. This is similar to what one finds for local $\Phi$-measurements in Sec. 4, where $\xi \sim \gamma^{-1/2}$ [see Eq. (46)]. In turn, for fixed $\gamma/\omega = 0.01$, the correlation length decays as $\xi \sim \eta_0^{-\beta}$ with $\beta = 0.226$, which is again comparable with local $\Phi$-measurements with $\xi \sim \eta_0^{-1/4}$. From the viewpoint of correlations functions, random measurements thus have a comparable effect as local $\Phi$-measurements. Both break the scale invariance of the Hamiltonian and induce a non-zero correlation length, which decreases proportional to the measurement strength.

## 6.3 Riccati Spectrum and Relaxation

Analogous to the non-zero correlation function $\xi$, random measurements also induce a non-zero relaxation rate in the dynamics towards the steady state. The Riccati spectrum for one realization of $m$ is displayed in Fig. 5(c), which reveals a gap in the real ($\kappa_0$) and imaginary parts ($\omega_0$) of the eigenvalues $\lambda_n$ [cf. Eq. (45)]. The behavior of both $\omega_0$ and $\kappa_0$ as a function of $\gamma$, and averaged over several realizations of $m$, is shown in Fig. 5(d). For small coupling constants $\gamma \ll \omega$, $\overline{\omega}_0 \sim \gamma^\delta$ with $\delta = 0.478$, which is consistent with the behavior of the correlation length and the identification $\omega_0 \sim \xi$. The average relaxation rate, however, scales as $\overline{\kappa}_0 \sim \gamma^\epsilon$ with $\epsilon = 0.968$. This is twice the value of the correlation length and therefore does not allow the identification $\overline{\kappa}_0 \sim \overline{\omega}_0 \sim \xi$, which we encountered in Eq. (46) of Sec. 4. Rather it yields $\overline{\kappa}_0 \sim \xi^2$, which corresponds to a dynamical critical exponent of $z = 2$, and to a classical (finite temperature) relaxation dynamics. It is thus consistent with the picture of relaxation into a highly excited state in the middle of the spectrum.

When increasing the measurement strength to values $\gamma \sim \omega$, the corresponding correlation length becomes smaller than the distance between two neighboring lattice sites, which also yields a breakdown of the scaling relations for $\overline{\omega}_0$ and $\overline{\kappa}_0$.

## 6.4 Volume Law Entanglement

We will now turn to the entanglement structure of the steady state with random measurements, which we characterize again in terms of the logarithmic negativity. In Fig. 5(e) we show the scaling of the half-system logarithmic negativity with system size $N$ for $\eta_0 = 1$. It reveals an extensive, volume law scaling of the entanglement,

$$\overline{\mathcal{N}_A[\boldsymbol{\sigma}]} \sim bN, \tag{65}$$

with linear scaling parameter $b$ and neglecting a constant offset. Volume law entanglement, combined with exponentially decaying correlations is a characteristic that is usually found in excited states. Here, it is fully attributed to the random measurements $\{O_l\}$.

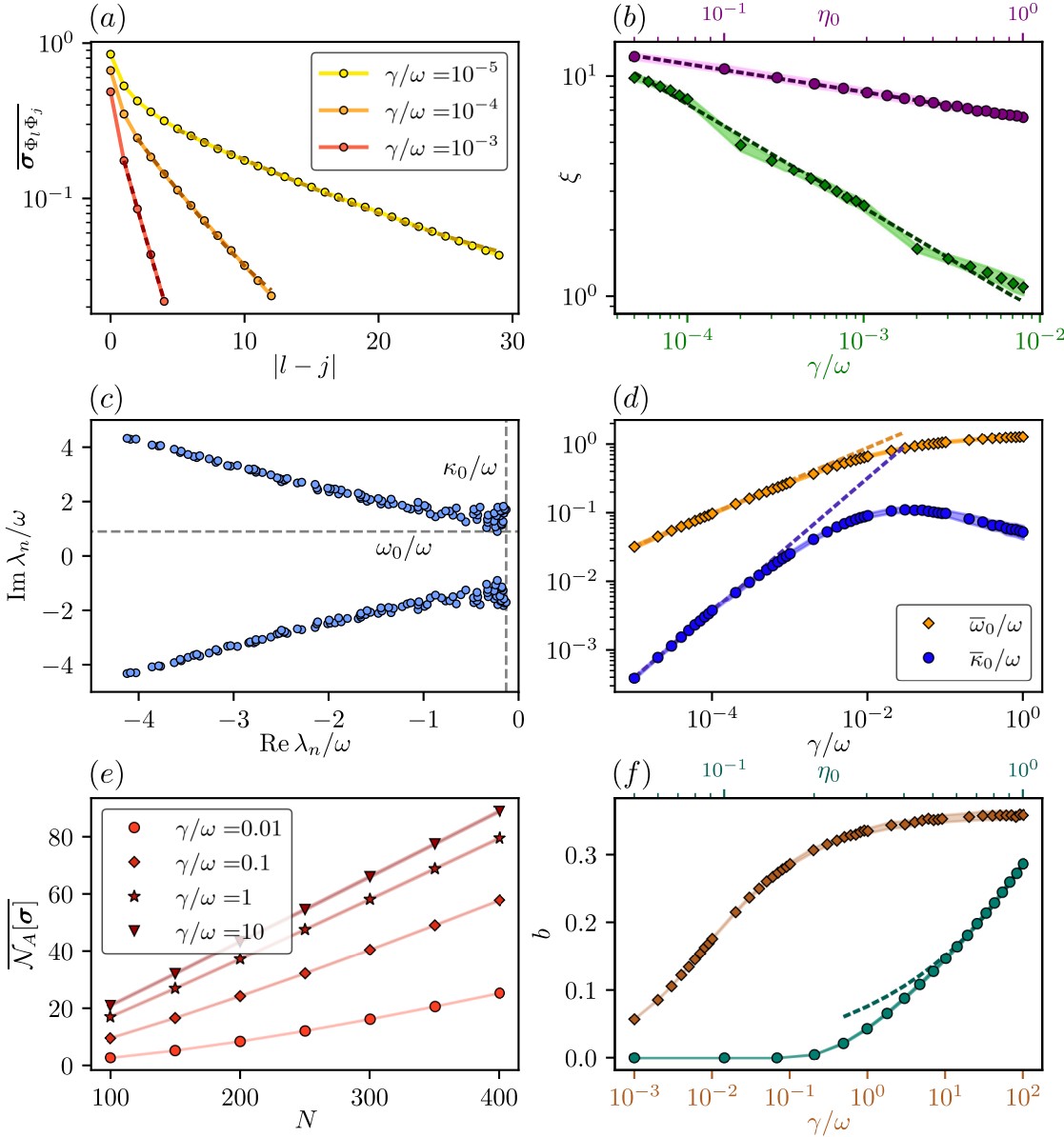

Figure 5: (a) Averaged real space correlations $\overline{\sigma_{\Phi_i \Phi_l}}$ for a system size of $N = 200$, $\eta_0 = 1$ and averaged over $n_{\text{sample}} = 500$ random measurement settings. (b) Scaling of the correlation length $\xi$ with increasing $\gamma$ (green markers) and $\eta_0$ (purple markers) for $N = 200$ and $n_{\text{sample}} = 500$. The shaded region corresponds to the variance between realizations. The systematic errors for the green line can be attributed to heuristics in finding the exponential tails (darker dashed lines) in (a). The green dashed line corresponds to the fit $\xi \sim \gamma^{-\alpha}$ with $\alpha = 0.478$, whereas the purple dashed line corresponds to $\xi \sim \eta_0^{-\beta}$ with $\beta = 0.226$. (c) Riccati spectrum for one realization of measurement operators with real (imaginary) gaps $\kappa_0$ ($\omega_0$). (d) Scaling of averaged $\overline{\omega}_0$ (orange markers) and $\overline{\kappa}_0$ (blue markers) with $\gamma$ for a system size of $N = 200$ and $n_{\text{sample}} = 500$. The orange (blue) dashed line corresponds to $\overline{\omega}_0 \sim \gamma^\delta$ with $\delta = 0.478$ ($\overline{\kappa}_0 \sim \gamma^\epsilon$ with $\epsilon = 0.968$). (e) Finite size scaling of half chain logarithmic negativity for half system averaged over $n_{\text{sample}} = 500$ realizations. (f) Dependenc of the linear scaling constant $b$ on $\gamma$ (brown) and $\eta_0$ (turquoise) obtained from the finite size scaling over system sizes from (e). The turquoise dashed line indicates a linear decay.

Varying the measurement strength $\gamma$ preserves the volume law structure [see Fig. 5(e)] but modifies the linear prefactor $b$, which is displayed in Fig. 5(f). The prefactor grows proportional to the measurement strength $\gamma$ for weak measurements $\gamma < \omega$ and then saturates around $\gamma \sim \omega$. This is around the value, at which the correlation length approaches the value of the lattice spacing, and when increasing $\gamma$ therefore no longer affects the steady state wave function. The monotonous growth of the entanglement entropy with $\gamma$ has also been observed for the critical setup in Sec. 5. In both cases the entanglement structure indicates that the measurements increase the non-locality of the wave function, which then leads to a growth of the entanglement entropy.

The volume law in the logarithmic negativity is robust against moving away from perfect measurements and reducing the measurement efficiency $\eta_0 < 1$. For sufficiently large values of $\eta_0 \approx 1$, the entanglement decreases linearly in $\eta_0$, i.e., $b \sim \eta_0$, see Fig. 5(f). When reaching smaller values of $\eta_0 \approx 0.5$, it starts to decrease faster than linear and obtains very small values, $b \approx 0$. The logarithmic negativity then approaches an area law.

In summary, fully randomized measurements $O_l = \sum_j m_{lj}\Phi_j$ lead to a steady state wave function (or density matrix), which mimics the behavior of a generic wave function in ergodic quantum mechanical systems: short-ranged correlations, finite temperature relaxation dynamics with a dynamical critical exponent $z = 2$, and a volume law entanglement structure. From this viewpoint, random measurement matrices in the Gaussian measurement framework act analogously to free random matrix Hamiltonians in isolated systems, and may be used as proxies for measurement-induced dynamics in generic systems.

# 7 Full Tomography of the Conditioned State

Among the key signatures that have been established for a theoretical characterization of measurement-induced phases and criticality in continuously or stroboscopically observed quantum systems, are the entanglement scaling or the timescale of purification of the conditioned quantum state. However, such quantities require in essence the knowledge of the full conditioned quantum many-body state, which is notoriously difficult, if yet possible to verify experimentally. The difficulty arises from both the exponential scaling of the number of measurements required for a full state tomography and the fact that we are interested in properties of the conditioned state $\rho_{c,t}$, which is the result of a specific sequence of random measurement outcomes. For discrete measurements, it will take exponentially many trials to re-obtain the very same measurement sequence and therefore to obtain multiple copies of $\rho_{c,t}$ to perform measurements on. For continuously monitored systems, the same trajectory of the measurement current will never appear twice. Thus, the question how to overcome this "post-selection barrier" in general and to detect measurement-induced criticality and phases in real experiments remains an open problem of utmost importance in this field [29, 68, 69].

We demonstrate in the following that also in this respect the free boson Kalman filter assumes a special role. For Gaussian measurements, the entanglement structure, the relaxation (and purification) dynamics, and the spatial correlations are encoded in the covariance matrix $\boldsymbol{\sigma}_{c,t}$. Not only can it be solved analytically, in this setting it is possible to reconstruct the conditioned density matrix $\rho_{c,t}$ and $\boldsymbol{\sigma}_{c,t}$ from the knowledge of the measurement outcomes and the measured observables. Thereby one can verify the very characteristics of measurement-induced phases discussed above. In order to demonstrate this, we will now concentrate on the actual aspect of a measurement, namely to learn something about the system at hand. In particular, we will describe a measurement protocol to determine conditioned expectation values and the full conditioned state $\rho_{c,t}$, which for Gaussian states is equivalent to the knowledge of $\boldsymbol{\sigma}_{c,t}$ and $\langle \boldsymbol{s} \rangle_{c,t}$ along a single measurement trajectory.

## 7.1 Observables and Tomography of the Unconditioned State

During each realization of a continuously measured trajectory, a stream of currents $I_t$ is recorded. In general, quantum mechanical observables are measured via the averaging of the measurement results over many experimental runs, and therefore we have to understand $\mathbb{E}(O) \simeq n^{-1} \sum_{i=1}^{n} O^{(i)}$ as a statistical average over $n \gg 1$ experimental runs, where $O^{(i)}$ denotes the outcome of the $i$-th experiment. In the same spirit, for a set of continuously measured operators $\{O_l\}$, one records the currents $\{I_{l,t}^{(i)}\}$, discussed in Eq. (12), in each experimental realization. Averaging the measured currents over many experimental runs then allows us to determine the unconditioned observables

$$\mathbb{E}(I_{l,t}) = \langle O_l \rangle_t = \mathrm{Tr}(O_l \rho_t). \tag{66}$$

Here, the state $\rho_t = \mathbb{E}\rho_{c,t} = e^{\mathcal{L}t}\rho_{t=0}$ evolves according to the master equation in Eq. (16) and corresponds to the unconditioned, trajectory averaged density matrix. Similarly, when averaging the fluctuations of the measured currents over many runs, we obtain

$$\mathbb{E}(I_{t+0^+} I_t^\top) = \frac{1}{4}\eta\Gamma M \Sigma_t M^\top \Gamma^\top \eta^\top. \tag{67}$$

The left of Eq. (67) is what is obtained in the experiment, which can then be used to compute the *unconnected correlator*

$$\Sigma_t = \frac{1}{2}\langle\{s, s^\top\}\rangle_t, \tag{68}$$

if the measurement operators $M$ and $\Gamma, \eta$ are know. The unconnected correlator corresponds to the expectation value with respect to the unconditioned density matrix $\langle\ldots\rangle_t = \mathrm{Tr}(\ldots\rho_t)$. The derivation of Eq. (67) is discussed in App. E.

Note that $\Sigma_t$ does not depend on a specific measurement record and therefore is *the same* for any large set of trajectories. This provides some freedom for implementing an experimental detection scheme. For instance, in order to determine $\Sigma_t$ for a given measurement setting, say $\Pi$-measurements, one can run the measurement-induced evolution exclusively with $\Pi$-measurements up to time $t - dt$. Then a complete set of measurements, i.e., of the $2N$ observables $\{O_l = \Phi_l, \Pi_l\}_{l=1}^N$, is implemented at time $t$ and this experiment is repeated $n \gg 1$ times. In this way we can determine the full $\Sigma_t$ at an arbitrary point in time. In a Gaussian setting, this corresponds to a full tomography of the unconditioned state, which can be done with a number of measurements that scales only polynomially in the system size and the evolution time. This information does not yet reveal any characteristic behavior of the measurement-induced dynamics, since unconditioned observables correspond to the limit of completely imperfect measurements, $\eta_0 = 0$. Nevertheless, the knowledge of $\Sigma_t$ will allow us to reconstruct the covariance matrix $\sigma_{c,t}$, which we demonstrate below.

## 7.2 Tomography of the Full Conditioned Density Matrix

The previous examples highlight that the standard method of obtaining expectation values from measurement results, via averaging over experimental measurement outcomes, only provide access to unconditioned expectation values and observables. However, the actual object of interest is the conditioned state of the system, $\rho_{c,t}$ (or equivalently $\sigma_{c,t}$ and $\langle s \rangle_{c,t}$), during a single experimental run. In order to extract $\sigma_{c,t}$ from the measured current $I_t$, we follow the protocol sketched in Fig. 1(c) and make use of Eq. (22), i.e., the deterministic (and therefore integrable) structure of the free boson Kalman filter. From this equation we obtain the identity

$$\sigma_{c,t} = \mathbb{E}(\sigma_{c,t}) = \Sigma_t - \mathbb{E}(\langle s \rangle_{c,t} \langle s \rangle_{c,t}^\top), \tag{69}$$

which shows the subtle interplay between conditioned and unconditioned means in the covariance matrix. This expression is remarkable in the regard that the only conditioned quantities the covariance matrix depends on are the first moments, which we will harness in the following. The first term, $\Sigma_t$, can be obtained in the 'standard' way from measurements of the averaged current fluctuations described above [see Eq. (67)]. The second term, $\mathbb{E}(\langle s \rangle_{c,t} \langle s \rangle_{c,t}^{\top})$, denotes the average over products of conditioned expectation values, which evolve according to Eq. (19). Substituting the noise increments through the measured currents, i.e.,

$$
\mathrm{d}\boldsymbol{W}_t = 4\Gamma^{-\frac{1}{2}}\eta^{\frac{1}{2}}\left(\boldsymbol{I}_t - \frac{\Gamma}{2}M\langle \boldsymbol{s} \rangle_{c,t}\right)\mathrm{d}t,
\tag{70}
$$

we obtain

$$
\langle \boldsymbol{s} \rangle_{c,t+\mathrm{d}t}^{(i)} = \langle \boldsymbol{s} \rangle_{c,t}^{(i)} + \left\{Jh\langle \boldsymbol{s} \rangle_{c,t}^{(i)} + 2\boldsymbol{\sigma}_{c,t}M^{\top}\eta\left(\boldsymbol{I}_t^{(i)} - \frac{\Gamma}{2}M\langle \boldsymbol{s} \rangle_{c,t}^{(i)}\right)\right\}\mathrm{d}t.
\tag{71}
$$

For a given measurement record $\boldsymbol{I}_t^{(i)}$, this equation describes a deterministic update of $\langle \boldsymbol{s} \rangle_{c,t}$. It, however, depends explicitly on the unknown $\boldsymbol{\sigma}_{c,t}$, and therefore, Eq. (71) and Eq. (69) must be solved iteratively. To do so one performs a large number of $n \gg 1$ experiments, each prepared in the same initial state with known values for $\langle \boldsymbol{s} \rangle_{c,t=0}$ and $\boldsymbol{\sigma}_{c,t=0}$. For each experiment the measurement outcomes $\{I_{l,t}^{(i)}\}$ are recorded and used to calculate the values of $\langle \boldsymbol{s} \rangle_{c,t+\mathrm{d}t}^{(i)}$ from the known values of $\langle \boldsymbol{s} \rangle_{c,t}^{(i)}$ and $\boldsymbol{\sigma}_{c,t}$. For the updated value of $\boldsymbol{\sigma}_{c,t+\mathrm{d}t}$ we incur the identity Eq. (69) with the predetermined values of the unconditioned correlations $\{\Sigma_\tau\}_{\tau \leq t}$ and the ensemble average performed over the set of $n$ conditioned first order moments $\langle \boldsymbol{s} \rangle_{c,t}^{(i)}$. With this, all the values for $\langle \boldsymbol{s} \rangle_{c,t}^{(i)}$ and $\boldsymbol{\sigma}_{c,t}$ can be constructed iteratively for a given measurement trajectory. This is equivalent to a reconstruction of the full conditioned state $\rho_{c,t}$ [see Eq. (18)], from which all other measurement-induced characteristics, such as the scaling of the entanglement, can be obtained.

Although the described procedure still requires many measurements and accurate knowledge about the experimental setup, it can be done with a polynomial number of measurements and thus overcomes the usual double exponential barrier encountered in the reconstruction of conditioned quantum many-body states. The reason for this surprising result is rooted in the fact that (i) Gaussians states can be described by a polynomial number of parameters that can be numerically integrated for rather large system sizes, and that (ii) due to the deterministic dynamics of the covariance matrix we do not have the problem of postselection. This means that we do not have to wait for an equivalent measurement pattern before the state tomography can be resumed.

Finally, let us point out that in many situations of interest, it might be enough to verify that the theoretically simulated estimate $\hat{\boldsymbol{\sigma}}_{c,t}$, which can be calculated directly from Eq. (20), is compatible with the obtained measurement results. In this case, the calculated value $\hat{\boldsymbol{\sigma}}_{c,t}$, and not the measured value of the covariance matrix, is used in Eq. (71) to evaluate the update for the first moments. Only at the final time, the covariance matrix $\boldsymbol{\sigma}_{c,t}^*$ is explicitly constructed from the ensemble average in Eq. (69). To do so, $\Sigma_t$ needs to be known only at a single point in time $t$, which substantially reduces the experimental cost. If $\|\boldsymbol{\sigma}_{c,t}^* - \hat{\boldsymbol{\sigma}}_{c,t}\| \ll 1$ we may conclude that $\boldsymbol{\sigma}_{c,t}^*, \hat{\boldsymbol{\sigma}}_{c,t} \to \boldsymbol{\sigma}_{c,t}$ almost surely.

## 8   Conclusion

We have demonstrated that bosonic Gaussian measurements represent a rather unique class of systems, whose conditioned dynamics under continuous measurements is exactly solvable.

Starting from the unitary free boson CFT, we examined three different types of continuous measurements, and found that the corresponding stationary states cover the three paradigmatic cases of area law, volume law, or a logarithmic scaling of the entanglement. Despite its simplicity, the Gaussian measurement framework thus represents an elementary model, which covers the conventional measurement-induced phases (volume and area law) and quantum criticality under measurements. Common for a Gaussian theory, it lacks the non-linear competition that leads to a measurement-induced phase transition between the different regimes. However, similar to Gaussian Hamiltonians (or Lindbladians), the Gaussian measurements capture the characteristics in each regime, and provide insights on and foster our understanding of the emergent behavior of more complicated, generic bosonic measurement dynamics.

In addition to the established classifier of measurement-induced dynamics, the entanglement structure of the steady state, we provide a set of additional observables. Both the spatial correlation functions and the relaxation behavior unambigously distinguish measurement-induced (or -enriched) criticality, and therefore conformally invariant behavior, from the gapped counterparts. This provides additional robust classifiers, which also work reliably for mixed state phases in the presence of imperfect measurements, or other environmental dephasing processes. The fact that all three observables are determined from the deterministic covariance matrix underpins the direct relation between correlations, relaxation behavior, and the entanglement structure in hybrid setups and points out their significance [37, 40, 41, 68, 74].

For the Gaussian measurements, the role of the covariance matrix is highlighted further by the fact that it can be partly (or completely) reconstructed on the basis of the already existing continuous measurement records. In a broader scope, this result shows that correlations between the measurement outcomes in a given continuously measured evolution protocol reveal information on the underlying wave function or density matrix. Statistical correlations between measurement outcomes at different times and (or) positions are nonlinear functions of the state of the system and thus are the type of quantities that contain nontrivial information. When this information is sufficient to distinguish different measurement-induced phases (as it is for Gaussian measurements), then this provides a robust experimental detection scheme.

Further directions, which may build up on the formalism and the results discussed here include the analysis of the apparent conformal invariance in the critical regime, and the impact of a dissipative environment on the hybrid dynamics. Conformal invariance is associated with both quantum and classical critical phenomena, and typically described by a universal ground or stationary state. While we observe scale invariant correlations and relaxation dynamics with universal exponents in the critical regime, the floating prefactor of the logarithmic entanglement growth indicates that some aspects of universality in the steady state are violated under measurements. Whether the hybrid dynamics yields indeed a quantum critical steady state, or only an apparently critical state, therefore is still an open question, which is relevant for the similar scenario of continuously observed fermions [37, 44, 74, 75].

Environmental dissipation, even if weak, will inevitably be present in experimental realizations of continuously measured evolution protocols. However, the interplay between measurements and dissipation is still barely understood, especially for dissipation induced by non-Hermitian Lindblad operators, such as particle gain and loss. Including such processes into the measurement-induced evolution is particularly relevant for the connection of measurement-induced phase transitions with quantum error correction [63, 64, 66], for which environmental dissipation represents a severe limitation [122–125]. Our work, with a simple but powerful formalism, may lay the foundation for this analysis of even more general boson models, which combine unitary evolution, measurements and dissipation.

# Acknowledgement

We thank S. Diehl, R. Küng, V. Eisert and B. Lapierre for fruitful discussions.

**Funding information** Y. M. and P. R. were supported by the Austrian Science Fund (FWF) through Grant No. P32299 (PHONED). M. B. acknowledges funding from the Deutsche Forschungsgemeinschaft (DFG, German Research Foundation) via grant DI 1745/2-1 under DFG SPP 1929 GiRyd.

# A   Ground State Covariance Matrix

The steady state covariance matrix in Eq. (55) is very similar to the ground-state covariance matrix in Eq. (8), which therefore has a prominent role in this work. In order to be self contained we restate this result, which can, among others, be found in [85]. The Hamiltonian is diagonalized by the symplectic transformation

$$H = \frac{\omega}{2} \sum_{ij} \Phi_i (\nu_N)_{ij} \Phi_j + \Pi_i \delta_{ij} \Pi_j = \frac{\omega}{2} \sum_n \epsilon_n^2 \Phi_n^2 + \Pi_n^2 = \frac{\omega}{2} \sum_n \epsilon_n (\tilde{\Phi}_n^2 + \tilde{\Pi}_n^2), \qquad (72)$$

where $\Phi_n = \sum_j O_{nj} \Phi_j$ and $\Pi_n = \sum_j O_{nj} \Pi_j$ with the same orthogonal matrix $O \in O(N)$. Here we used the fact that position and momentum operators are uncoupled, such that the symplectic transformation $S \in \mathrm{Sp}(2N, \mathbb{R})$, which diagonalized $H$, simplifies to $S = O \oplus O$. $\nu_N$ is diagonalized by these orthogonal transformations with eigenvalues $\{\epsilon_n^2\}$. Further, we rescaled $\tilde{\Phi}_n = \Phi_n / \sqrt{\epsilon_n}$ and $\tilde{\Pi}_n = \sqrt{\epsilon_n} \Pi_n$ with $\langle \tilde{\Pi}_n^2 \rangle_{\mathrm{GS}} = \epsilon_n / 2$ and $\langle \tilde{\Phi}_n^2 \rangle_{\mathrm{GS}} = 1/(2\epsilon_n)$. Therefore, the real space correlations in the ground state are

$$\sigma_{\Phi_i \Phi_j} = \langle \Phi_i \Phi_j \rangle_{\mathrm{GS}} = \frac{1}{2} \sum_n O_{in} \epsilon_n^{-1} O_{nj} = \frac{1}{2} (\nu_N^{-\frac{1}{2}})_{ij} \qquad (73)$$

and the momentum correlations are found in an analogous way. This gives the expression in Eq. (8).

# B   Gaussian Measurements and Stochastic Schrödinger Equation

## B.1   Measurement Results Follow a Wiener Process

In the main text we claimed in Eq. (12) that the measurement outcomes $I_t$ are distributed in a Gaussian way around the mean $\langle O \rangle_{c,t}$. In the following we will briefly sketch the derivation of this result. We start by considering a representation of the trace $\mathrm{Tr}(\dots) = \int do \langle o| \dots |o \rangle$ in terms of the eigenvectors $O|o\rangle = o|o\rangle$. The probability for outcome $I_t$ is then expressed as

$$\begin{aligned}
\Pr(I_t) = \mathrm{tr}\left( E_{I_t}^\dagger E_{I_t} \rho_{c,t} \right) &= \left( \frac{8 dt}{\pi \gamma} \right)^{\frac{1}{2}} \int do \, \langle o| \rho_{c,t} |o \rangle \exp\left[ -\frac{8 dt}{\gamma} \left( I_t - \frac{\gamma}{2} o \right)^2 \right] \\
&\simeq \left( \frac{8 dt}{\pi \gamma} \right)^{\frac{1}{2}} \int do \, \delta(o - \langle O \rangle_{c,t}) \exp\left[ -\frac{8 dt}{\gamma} \left( I_t - \frac{\gamma}{2} o \right)^2 \right] \qquad (74) \\
&= \left( \frac{8 dt}{\pi \gamma} \right)^{\frac{1}{2}} \exp\left[ -\frac{8 dt}{\gamma} \left( I_t - \frac{\gamma}{2} \langle O \rangle_{c,t} \right)^2 \right].
\end{aligned}$$

From the first to the second line we used that for small time increments $dt$ with $(\gamma dt)^{-1} \gg \Delta O_t^2$, where $\Delta O_t^2 = \langle O^2 \rangle_{c,t} - \langle O \rangle_{c,t}^2$ denotes the width of $\langle o | \rho_{c,t} | o \rangle$, the Gaussian measurement profile is much wider than the wave function, such that we approximate the latter to be delta-localized. From this it follows that the measurement results are distributed in a Gaussian way around the mean $\langle O \rangle_{c,t}$ and therefore the measurement results follow the Wiener process from Eq. (12) in the main text.

## B.2 Derivation of the Stochastic Schrödinger Equation

In Eq. (9) the SSE was introduced and subsequently its derivation was sketched. Here we explicitly spell out the crucial steps of the derivation, which we omitted in the main text. This presentation is inspired by [79, 81, 90]. The starting point is the state update in Eq. (11). Expanding the measurement operator to lowest order in $dt$, using that Eq. (12) is a Wiener process and applying the Itō rule $dW_t^2 = dt$ we obtain

$$E_{I_t} |\psi_t\rangle_c \simeq \left(1 - iHdt - \frac{\gamma}{2}(O^2 - 4O\langle O \rangle_{c,t})dt + \sqrt{\gamma}OdW_t\right)|\psi_t\rangle_c, \tag{75}$$

where we neglected terms of order $O(dt^{3/2})$. The norm of Eq. (75) to lowest order is

$$\|E_{I_t} |\psi_t\rangle_c\| \simeq 1 + 2\sqrt{\gamma}\langle O \rangle_{c,t}dW_t + 4\gamma\langle O \rangle_{c,t}^2 dt \tag{76}$$

and inverting this expression with a subsequent expansion yields

$$\|E_{I_t} |\psi_t\rangle_c\|^{-1} = 1 - \frac{\gamma}{2}\langle O \rangle_{c,t}^2 dt - \sqrt{\gamma}\langle O \rangle_{c,t}dW_t \tag{77}$$

to lowest order in $dt$. Combining Eq. (77) and Eq. (75) we obtain the discrete stochastic Schrödinger equation

$$|\psi_{t+dt}\rangle_c \simeq \left(1 - i\left(H - i\frac{\gamma}{2}\left(O - \langle O \rangle_{c,t}\right)^2\right)dt + \sqrt{\gamma}(O - \langle O \rangle_{c,t})dW_t\right)|\psi_t\rangle_c, \tag{78}$$

and from $d|\psi_t\rangle_c = |\psi_{t+dt}\rangle_c - |\psi_t\rangle_c$ we obtain the expression in Eq. (9) for a single measurement. This derivation is straightforwardly extended to the simultaneous measurement of multiple observables $O_i$ with strengths $\gamma_i$ and independent outcomes $\{I_{i,t}\}$.

# C Details on the Kalman Filter

## C.1 Derivation of the Evolution Equation for the Conditioned First Moments and the Covariance Matrix

In this section we derive the equations of motion for the first moments and the covariance matrix from Eq. (19) and Eq. (20) of the main text. From the SME in Eq. (15) we derive the equation of motion for a generic operator $A$, which becomes

$$d\langle A \rangle_{c,t} = -\left(i\langle [A, H] \rangle_{c,t}dt + \sum_i \frac{\gamma_i}{2}\langle [O_i, [O_i, A]] \rangle_{c,t}\right)dt + \sum_i \sqrt{\eta_i\gamma_i}\langle \{O_i, A\} - \langle O_i \rangle_{c,t}A \rangle_{c,t}dW_{j,t}. \tag{79}$$

We first start by considering the unitary limit $\gamma_i = 0$. In order to examine the unitary evolution of the first moments we start by considering the commutator

$$\dot{s}_i = -i[s_i, H] = -\frac{i}{2}\sum_{a,b} h_{ab}[s_i, s_a s_b] = -\frac{i}{2}\sum_{a,b} iJ_{ia}h_{ab}s_b + iJ_{ib}h_{ba}s_a) = (Jh\mathbf{s})_i, \tag{80}$$

from which Eq. (6) follows. The equations of motion of the operators centered around the mean, $\tilde{s}_i = s_i - \langle s_i \rangle$, are the same. From this the equation of motion for the unitary evolution of the covariance matrix, $\dot{\boldsymbol{\sigma}}_{ij} = (\langle \{\dot{\tilde{s}}_i, \tilde{s}_j\}\rangle + \langle \{\tilde{s}_i, \dot{\tilde{s}}_j\}\rangle)/2$, is found to be Eq. (7) from the main text. Let us now turn to $\gamma_i > 0$, where we start by considering the first moments with the last term in Eq. (79), which becomes

$$\sum_i \sqrt{\eta_i \gamma_i} \langle \{O_i, s_j\} - 2\langle O_i \rangle_{c,t} s_j \rangle_{c,t} \mathrm{d}W_{i,t} = \sum_i \sqrt{\eta_i \gamma_i} M_{ik} (\langle s_k s_j \rangle_{c,t} + \langle s_j s_k \rangle_{c,t} - 2\langle s_j \rangle_{c,t} \langle s_k \rangle_{c,t}) \mathrm{d}W_{i,t}$$
$$= (2\boldsymbol{\sigma}_{c,t} M^\top \sqrt{\eta\Gamma} \mathrm{d}\boldsymbol{W}_t)_j. \tag{81}$$

The double commutator $[s_i, [s_i, s_j]] = 0$ from the second term in Eq. (79) vanishes and therefore does not contribute to the final result of Eq. (19) in the main text. We will now turn to the equation of motion for the covariances. First of all we note that the equation of motion for the first moment is now a stochastic differential equation of the Itō kind. From this we obtain the evolution of the covariances from

$$\mathrm{d}(\boldsymbol{\sigma}_{c,t})_{ij} = \frac{1}{2}(\langle \{\mathrm{d}s_i, s_j\}\rangle_{c,t} + \langle \{s_i, \mathrm{d}s_j\}\rangle_{c,t}) - \langle s_i \rangle_{c,t} \mathrm{d}\langle s_j \rangle_{c,t} - \mathrm{d}\langle s_i \rangle_{c,t} \langle s_j \rangle_{c,t} - \mathrm{d}\langle s_i \rangle_{c,t} \mathrm{d}\langle s_j \rangle_{c,t}$$
$$= \frac{1}{2}(\langle \{\mathrm{d}\tilde{s}_i, \tilde{s}_j\}\rangle_{c,t} + \langle \{\tilde{s}_i, \mathrm{d}\tilde{s}_j\}\rangle_{c,t}) - \mathrm{d}\langle s_i \rangle_{c,t} \mathrm{d}\langle s_j \rangle_{c,t}, \tag{82}$$

where the last term in both lines is attributed to the Itō correction to the differential increment, $\mathrm{d}(fg) = \mathrm{d}f\,g + f\,\mathrm{d}g + \mathrm{d}f\,\mathrm{d}g$. The unitary evolution will be unaltered, but there will be a correction attributed to the double commutator, which we obtain by considering

$$\sum_i \frac{\gamma_i}{2}\langle [O_i, [O_i, \tilde{s}_j \tilde{s}_k]]\rangle_{c,t} = -\sum_{i,m,n} \frac{\gamma_i}{2} M_{im} M_{in} \langle [s_m, [s_n, \tilde{s}_j \tilde{s}_k]]\rangle_{c,t}$$
$$= \sum_{i,m,n} \frac{\gamma_i}{2} M_{im} M_{in} (J_{nj}J_{mk} + J_{nk}J_{mj}) = (JM^\top \Gamma MJ^\top)_{jk}, \tag{83}$$

which gives the inhomogeneous contribution in Eq. (20). The most important feature of the linear measurements is that the stochastic contribution in the equation of motion for the co-variances cancels. This is seen by noting that $\langle \tilde{s}_i \rangle = 0$ and since the state is at all times in the manifold of Gaussian states, Wick's theorem applies. Therefore, the prefactor in the stochastic contribution in Eq. (79),

$$\langle \{s_i, \tilde{s}_j \tilde{s}_k\} - \langle s_i \rangle_{c,t} \tilde{s}_j \tilde{s}_k \rangle_{c,t} = \langle s_i \tilde{s}_j \tilde{s}_k \rangle_{c,t} + \langle \tilde{s}_j \tilde{s}_k s_i \rangle_{c,t} - 2\langle s_i \rangle_{c,t} \langle \tilde{s}_j \tilde{s}_k \rangle_{c,t}$$
$$= \langle s_i \tilde{s}_j \rangle_{c,t} \langle \tilde{s}_k \rangle_{c,t} + \langle s_i \tilde{s}_k \rangle_{c,t} \langle \tilde{s}_j \rangle_{c,t} + \langle \tilde{s}_j s_i \rangle_{c,t} \langle \tilde{s}_k \rangle_{c,t} + \langle \tilde{s}_k s_i \rangle_{c,t} \langle \tilde{s}_j \rangle_{c,t}$$
$$+ \langle s_i \rangle_{c,t} \langle \tilde{s}_j \tilde{s}_k \rangle_{c,t} + \langle \tilde{s}_j \tilde{s}_k \rangle_{c,t} \langle s_i \rangle_{c,t} - 2\langle s_i \rangle_{c,t} \langle \tilde{s}_j \tilde{s}_k \rangle_{c,t} = 0, \tag{84}$$

cancels. Finally, using the Itō rule we see that

$$\mathrm{d}\langle \boldsymbol{s} \rangle_{c,t} \mathrm{d}\langle \boldsymbol{s} \rangle_{c,t}^\top = 4\boldsymbol{\sigma}_{c,t} M^\top \eta\Gamma M \boldsymbol{\sigma}_{c,t} + O(\mathrm{d}t^{3/2}), \tag{85}$$

which concludes the derivation of Eq. (20).

## C.2 Alternative Argument for the Deterministic Dynamics of Covariances

In App. C.1 we showed that the evolution of the first moments is noisy, whereas from Wick's theorem it follows that the dynamics of the covariance matrix is deterministic. In the following we will present an alternative argument for this remarkable property of Gaussian measurements. We start with the measurement operator $O = a\Phi + b\Pi = m^\top \boldsymbol{s}$ focusing on a single site

for simplicity. As above we defined $s = (\Phi, \Pi)^\top$ and $m = (a, b)^\top$. The measurement operator Eq. (10) becomes

$$E_{I_t} \sim \exp\left(-\frac{1}{2}\mathrm{d}t(s - j_t)^\top Q(s - j_t)\right), \tag{86}$$

where we have defined the quadratic form

$$Q = 2\gamma m\, m^\top \qquad \text{and the noisy vector} \qquad j_t = \gamma^{-1}\begin{pmatrix} a^{-1} \\ b^{-1} \end{pmatrix} I_t. \tag{87}$$

Note that $Q$ is deterministic while only $j_t$ depends on the stochastic measurement outcomes. The operator $E_{I_t}$ is decomposed in terms of an imaginary time evolution $U_{\mathrm{d}t} = e^{-iJQ\mathrm{d}t}$ and the displacement operator $\mathcal{D}(v) = e^{iv^\top Js}$ as

$$E_{I_t} \sim \mathcal{D}(-j_t)\mathcal{D}(U_{\mathrm{d}t}^{-1}j_t)e^{-\frac{1}{2}s^\top Qs\,\mathrm{d}t}. \tag{88}$$

Applying the measurement operator the state at time $t$ will give the updated first moments and covariance matrix for one time step $\mathrm{d}t$:

$$\langle s \rangle_{c,t} \quad \rightarrow \quad \langle s \rangle_{c,t+\mathrm{d}t} \sim \langle s \rangle_{c,t} + (U_{\mathrm{d}t}^{-1} - \mathbb{1}_2)j_t, \tag{89}$$

$$\sigma_{c,t} \quad \rightarrow \quad \sigma_{c,t+\mathrm{d}t} \sim U_{\mathrm{d}t}\, \sigma_{c,t}\, U_{\mathrm{d}t}^\top, \tag{90}$$

where we neglected normalization constants. The first moments are updated in a stochastic way according to the outcomes of $I_t$, while the update of the covariance matrix is deterministic. This is not too surprising since we know from unitary time evolution that Hamiltonian terms linear in the fields will only affect the dynamics of the first moments.

## C.3   Review of the Classical Kalman Filter

In Sec. 3.1 we introduced the notion of the free boson Kalman filter to denote the conditioned equations of motion for the moments. In the following we will motivate this notation by first reviewing the classical Kalman filter and then making the connection to the quantum problem below.

In classical control theory, or more specifically in classical estimation theory, one frequently encounters the following problem: The state of a system of interest is described by the real vector $x_t = (x_{1,t}, \ldots, x_{N,t})^\top$ and obeys the following linear equation of motion

$$\dot{x}_t \quad = \quad Ax_t + V_x\zeta_t. \tag{91}$$

Its deterministic part is described by the $N \times N$-matrix $A$, but the system is also affected by the *process noise* characterized by $\mathbb{E}\zeta_{i,t} = 0$ and $\mathbb{E}\zeta_{i,t}\zeta_{j,s} = (V_xV_x^\top)_{ij}\delta(t-s)$. The state is continuously monitored through a noisy measurement device with outcome

$$y_t \quad = \quad Cx_t + V_y\xi_t. \tag{92}$$

Here, $C$ is an $N_m \times N$ matrix, where $N_m$ is the number of independent measurements, and the *measurement noise* is characterized by $\mathbb{E}\xi_{i,t} = 0$ and $\mathbb{E}\xi_{i,t}\xi_{j,s} = (V_yV_y^\top)_{ij}\delta(t-s)$. There are no cross-correlations, $\mathbb{E}\xi_{i,t}\zeta_{j,s} = 0$.

The problem of *filtering* in classical estimation theory [98] refers to the problem of finding an estimator $\hat{x}_t$ provided that the measurement results $\{y_t\}$ and some aspects of the system dynamics are known. This estimator is chosen such that it converges in some sense to the true unknown system state $x_t$. One prominent example, where it is assumed that the system dynamics $A$, as well as the strengths of the process and measurement noises ($V_x$ and $V_y$) are

known, is the *Kalman filter*. It is the prescription for constructing an optimal estimator such that

$$J_t = \mathbb{E}\,\boldsymbol{\epsilon}_t^\top \boldsymbol{\epsilon}_t \longrightarrow \min. \qquad \text{with} \qquad \boldsymbol{\epsilon}_t \equiv \boldsymbol{x}_t - \hat{\boldsymbol{x}}_t. \tag{93}$$

This means that the fluctuations in $\boldsymbol{\epsilon}_t$, which is the deviation of the estimator from the true state, are minimized. In this case the Kalman-filter can be treated analytically and the equation of motion for the estimated state $\hat{\boldsymbol{x}}_t$ evolves according to the equation of motion

$$\dot{\hat{\boldsymbol{x}}}_t \;=\; A\hat{\boldsymbol{x}}_t + K_F(\boldsymbol{y}_t - \hat{\boldsymbol{y}}_t), \tag{94}$$

where $\hat{\boldsymbol{y}}_t = C\hat{\boldsymbol{x}}_t$ are the estimated measurement results, consistent with the estimator dynamics. The usual procedure in filtering is the following; an experimentalist who continuously receives a stream of measurement results $\boldsymbol{y}_t$ uses Eq. (94) to solve for the estimator $\hat{\boldsymbol{x}}_t$ and thereby approximates the true system state $\boldsymbol{x}_t$. The Kalman filter refers to the choice of the *gain*, $K_F$, such that the cost function $J_t$ is minimized. The simplicity of the Kalman filter permits that the gain $K_F$ can be obtained from a simple variational computation [98] and has the explicit form

$$K_F = \boldsymbol{S}_t C^\top (V_I V_I^\top)^{-1} \qquad \text{with} \qquad \boldsymbol{S}_t = \mathbb{E}\,\epsilon_t \epsilon_t^\top, \tag{95}$$

which is the covariance of the estimation error, which has to satisfy the Riccati differential equation

$$\dot{\boldsymbol{S}}_t = A\boldsymbol{S}_t + \boldsymbol{S}_t A^\top + V_x V_x^\top - \boldsymbol{S}_t C^\top (V_y V_y^\top)^{-1} C\boldsymbol{S}_t. \tag{96}$$

The system state is referred to as *observable* if the dynamics of the estimator is fully under control by the choice of the matrix $K_F$. This means that the eigenvalues of $A - K_F C$ can be tuned to be all negative by the choice of $K_F$ alone.

## C.4   Connection between Classical and Quantum Kalman Filter

After this brief review of classical estimation theory and the Kalman filter, we are now returning to the quantum mechanical case of conditioned dynamics. Before proceeding we make the disclaimer that duality to the classical Kalman filter can be made more explicit and rigorous within the notion of *quantum-filtering* and *-probability*, which we will avoid here and refer the reader to [91, 92].

The formal connection between the classical Kalman filter and the continuously monitored quantum dynamics can be made by making the following identification in the Riccati equation of the classical Kalman filter in Eq. (96) with quantities from the quantum mechanical problem: The covariance matrix of the estimation errors is identified with the covariance matrix, $\boldsymbol{S}_t \to \boldsymbol{\sigma}_{c,t}$, the system dynamics with the unitary Hamiltonian dynamics, $A \to Jh$, the accessible points for the measurement with the measurement matrix, $C \to \Gamma M/2$, and finally the process and measurement noise, $V_x \to V_s = J^\top M \Gamma^{\frac{1}{2}}$ and $V_y \to V_I = 4\sqrt{\eta/\Gamma}$, respectively. With these replacements the Riccati equation for the estimation error $\boldsymbol{S}_t$ becomes identical to the equation of motion for the covariance matrix $\boldsymbol{\sigma}_{c,t}$ in Eq. (20) under continuous observation. Identifying the estimator with conditioned first moments, $\hat{\boldsymbol{x}}_t \to \langle \boldsymbol{s} \rangle_{c,t}$, in the equations of motion for the estimator, Eq. (94), as well as the above replacements in the Kalman gain $K_F$, we obtain the noisy equations of motion for the first moments we encountered in Eq. (19). The estimated measurement is identified with $\hat{\boldsymbol{y}}_t \to \hat{\boldsymbol{I}}_t = \Gamma M \langle \boldsymbol{s} \rangle_{c,t}/2$ in the quantum mechanical case. The conditioned state of the system $\rho_{c,t}$, which is governed by the stochastic master equation can therefore be understood as the estimator of the true state of the system when all the measured results are used. From this point of view, the Wiener noise in Eq. (12) has to be understood as being obtained from

$$\mathrm{d}\boldsymbol{W}_t = V_I^{-1}\left(\boldsymbol{I}_t - \frac{\Gamma}{2}M\langle\boldsymbol{s}\rangle_{c,t}\right)\mathrm{d}t = V_I^{-1}(\boldsymbol{I}_t - \hat{\boldsymbol{I}}_t)\mathrm{d}t, \tag{97}$$

where $V_I = \frac{1}{4}\eta^{-\frac{1}{2}}\Gamma^{\frac{1}{2}}$. The SME in this case reduces to

$$
\begin{aligned}
\dot{\rho}_{c,t} = & -i[H, \rho_{c,t}] - \sum_j \frac{\gamma_j}{2}[O_j, [O_j, \rho_{c,t}]] \\
& + \sum_j \sqrt{\eta_j \gamma_j}\big[\{O_j, \rho_{c,t}\} - 2\langle O_j\rangle_{c,t}\rho_{c,t})(V_I^{-1})_{jj}(I_{j,t} - \hat{I}_{j,t})\big].
\end{aligned}
\tag{98}
$$

It is reassuring that a continuously measured quantum state, updated by the measurement results, and completely derived within the framework of quantum mechanics, reduces to the well known classical filtering equations for Gaussian measurements.

# D   Solution of the Riccati Equation

In this section we present the general solution of the algebraic Riccati equation (23). This is crucial for the solution of quadratic matrix equations arising for instance in Eq. (34) and it is thus the key to solving for the steady states from the algebraic Riccati equation. We start by rewriting this quadratic matrix equation in a more general form

$$
-X^\top A X + X^\top B + B^\top X + C = 0.
\tag{99}
$$

Here $A > 0$ is a real symmetric non-negative $n \times n$ matrix, which is in our case always fulfilled since it has the form $A = M^\top \eta \Gamma M \geq 0$ and therefore also $A^\top = A$ follows. This is crucial for the existence of a unique solution [126]. The inhomogeneity $C = C^\top$ is a real symmetric $n \times n$-matrix, which is clearly satisfied for our choice of $C = J^\top M^\top \eta \Gamma M J$. The $n \times n$ matrices $B$ and $X$ are real.

Although in the this work $A$ is invertible, we will consider a slightly more general situation, where it can in principle be non-invertible and therefore we introduce the Moore-Penrose pseudo-inverse $A^+$ with $A^+ A = A A^+ = \mathbb{1}_n$, which also commutes with the transposition $(A^+)^\top = A^+$ [126]. It is now possible to complete the square and simplify Eq. (99) to

$$
-Y^\top A Y + D = 0 \qquad \text{where} \quad Y = X - A^+ B \quad \text{and} \quad D = B^\top A^+ B + C.
\tag{100}
$$

Since $A$ is non-negative we can define $U = A^{\frac{1}{2}} Y$ such that Eq. (100) becomes

$$
U^\top U = D \qquad \text{with the solution} \qquad U = \pm Q D^{\frac{1}{2}}.
\tag{101}
$$

This solution is well defined due to $D \geq 0$, which is fulfilled in the above situation. Furthermore we defined the orthogonal matrix $Q \in O(n)$, which was in our case taken to be $Q = \mathbb{1}_n$. Using the Moore-Penrose pseudo-inverse of the matrix square-root $A^{\frac{1}{2}+}$ we obtain the general solution

$$
Y = \pm A^{\frac{1}{2}+} Q D^{\frac{1}{2}} + (\mathbb{1}_n - B^{\frac{1}{2}+} B^{\frac{1}{2}}) W
\tag{102}
$$

with an arbitrary real $n \times n$ matrix $W$ and eventually we obtain the solution

$$
X = A^+ B \pm A^{\frac{1}{2}+} Q (B^\top A^+ B + C)^{\frac{1}{2}} + (\mathbb{1}_N - B^{\frac{1}{2}+} B^{\frac{1}{2}}) W.
\tag{103}
$$

Note that in the special cases discussed above $B^+ = B^{-1}$ and therefore the second part of the above solution was neglected.

# E  Unconditioned Observables from Averaging

In this section we will review the derivation of Eq. (67), a standard result from quantum optics [78,81] in the formalism we established above. We will start from the stochastic master equation (15) and the measurement outcomes (currents) from Eq. (10)

$$d\rho_{c,t} = (\mathcal{L}dt + \sum_l \sqrt{\eta_l \gamma_l}\, dW_{l,t}\mathcal{H}[O_l])\rho_{c,t} \qquad I_{j,t} = \frac{\gamma_j}{2}\langle O_j\rangle_{c,t} + V_{I,j}\xi_{j,t}, \tag{104}$$

where we introduced the shorter notation $V_{I,j} = \frac{1}{4}\sqrt{\frac{\gamma_j}{\eta_j}}$ and we remind that $\xi_{j,t} = \frac{dW_{j,t}}{dt}$. We furthermore use the notation

$$\mathcal{L}\rho_{c,t} = -i[H,\rho_{c,t}] - \sum_l \frac{\gamma_l}{2}[O_l,[O_l,\rho_{c,t}]], \qquad \mathcal{H}[O_l]\rho_{c,t} = O_l\rho_{c,t} + \rho_{c,t}O_l - 2\langle O_l\rangle_{c,t}\rho_{c,t}. \tag{105}$$

When evaluating the equal time current-current correlation functions, it is more favorable to point-split the correlator $\mathbb{E}I_{i,t}I_{j,t} = \lim_{dt\to 0^+}\mathbb{E}I_{i,t+dt}I_{j,t}$. Multiplying out this correlator we obtain

$$\mathbb{E}I_{i,t+dt}I_{j,t} = \frac{\gamma_i\gamma_j}{4}\mathbb{E}\langle O_i\rangle_{c,t+dt}\langle O_j\rangle_{c,t} + \frac{\gamma_i}{2}V_{j,I}\mathbb{E}\langle O_i\rangle_{c,t+dt}\xi_{j,t} + V_{i,I}\frac{\gamma_j}{2}\underbrace{\mathbb{E}\xi_{i,t+dt}\langle O_j\rangle_{c,t}}_{=0}$$
$$+ \delta_{ij}V_{I,i}V_{I,j}\delta(dt). \tag{106}$$

In the third term $\langle O_j\rangle_{c,t}$ only depends on $\xi_{\tau<t}$ and due to the white-noise property the expectation value vanishes, $\mathbb{E}\xi_{i,t+\tau}\xi_{j,t} = \delta_{ij}\delta(\tau)$. The fourth term vanishes due to the same reason. The second term requires more care and we start by writing it out explicitly

$$\frac{\gamma_i}{2}V_{j,I}\mathbb{E}\langle O_i\rangle_{c,t+dt}\xi_{j,t} = \frac{\gamma_i}{2}V_{I,j}\mathbb{E}\,\mathrm{Tr}\left(O_i\left(e^{\mathcal{L}dt} + \sum_l \sqrt{\eta_l\gamma_l}\,dW_{l,t}\mathcal{H}[O_l]\right)\rho_{c,t}\right)\frac{dW_{j,t}}{dt}, \tag{107}$$

where we used

$$\rho_{c,t+dt} \simeq (e^{\mathcal{L}dt} + \sum_l \sqrt{\eta_l\gamma_l}dW_{l,t}\mathcal{H}[O_l])\rho_{c,t}. \tag{108}$$

Using the Itō rule $dW_{i,t}dW_{j,t} = \delta_{ij}dt$ and the vanishing mean $\mathbb{E}\xi_{i,t} = 0$ of a white-noise process, we obtain

$$\frac{\gamma_i}{2}V_{I,j}\mathbb{E}\langle O_i\rangle_{c,t+dt}\xi_{j,t} = \frac{1}{2}\frac{\gamma_i\gamma_j}{4}\mathbb{E}(O_i(O_j\rho_{c,t} + \rho_{c,t}O_j - 2\langle O_j\rangle_{c,t}\rho_{c,t}))$$
$$= \frac{\gamma_i\gamma_j}{4}\left(\frac{1}{2}\mathbb{E}\langle\{O_i,O_j\}\rangle_{c,t} - \mathbb{E}\langle O_i\rangle_{c,t}\langle O_j\rangle_{c,t}\right), \tag{109}$$

with the anti-commutator $\{A,B\} = AB + BA$. Plugging this expression back into Eq. (106) we obtain

$$\mathbb{E}I_{i,t+dt}I_{j,t} = \frac{\gamma_i\gamma_j}{4}\left(\mathbb{E}\langle O_i\rangle_{c,t+dt}\langle O_j\rangle_{c,t} - \mathbb{E}\langle O_i\rangle_{c,t}\langle O_j\rangle_{c,t} + \frac{1}{2}\mathbb{E}\langle\{O_i,O_j\}\rangle_{c,t}\right). \tag{110}$$

When taking the limit $dt \to 0^+$ the term $\delta(0^+) = 0$ and the first two terms cancel and we are left with the unconditioned fluctutations

$$\mathbb{E}I_{i,t+0^+}I_{j,t} = \frac{\gamma_i\gamma_j}{4}\frac{1}{2}\langle\{O_i,O_j\}\rangle_t, \tag{111}$$

where we used $\mathbb{E}\langle\ldots\rangle_{c,t} = \mathrm{Tr}(\ldots\rho_t) = \langle\ldots\rangle_t$. Rewriting $O_i = \sum_m M_{im}s_m$ we obtain Eq. (67) from the main text.

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
