# Peer review of "Continuous Gaussian Measurements of the Free Boson CFT: An Exactly Solvable Model"

_SciPost Physics_

## Round 1 · Referee Report · Anonymous (Referee 1) · 2021-10-13

Strengths

The paper: 1. is self-consistent, clear and very well-written; 2. provides relevant and timely results on the field of many-body hybrid unitary-measurement dynamics.

Report

The Authors introduce a class of models where the system is unitarily evolved through a bosonic CFT and monitored continuously by linear self-adjoint bosonic operators. They consider two setups where the measurement record is either perfect or imperfect, and they provide exact and analytical solutions to the problem in both setups. They investigate purity, relaxation times, correlation functions, and entanglement. Since the state may be mixed (as in the case of imperfect measurement records) they use logarithmic negativity to ensure they measure only quantum correlations.
The Authors show that the choice of measurement (type of operators involved, and range of measurement) is fundamental in the steady-state property of the system. They finally show how full tomography is achievable for these states, with consequences in potential experimental implementations.

I enjoyed reading this paper, which I find very well-written, clear, and relevant and therefore I recommend it for publication. Having an analytically solvable model in the field of many-body monitored quantum dynamics is rare, as usually, the measurements introduce stochasticity in the system. Previous work has shown how the conditional average of an observable can be tackled by replica symmetry techniques; here remarkably the exact solution is given at a level of trajectory, provided the observable of interest is a function of the correlation matrix (second cumulant) $\boldsymbol{\sigma}$. In fact, the measurement record is relevant only for the first cumulant, which is required for the full tomography of the state.
An important discussion is one regarding the imperfect measurements. To the best of my knowledge, this is the first contribution to treat in the contest of many-body monitored dynamics the imperfect measurements. The Authors show that, even in the measurement-enriched critical phase, this imperfect track record only enters as a mild correction to the entanglement scaling properties (but preserving the logarithmic behavior).
Overall the ideas within this contribution allow investigating fundamental properties of the interplay between measurements and unitary dynamics.
  • validity: top
  • significance: top
  • originality: high
  • clarity: top
  • formatting: perfect
  • grammar: excellent

Author:  Yuri Minoguchi  on 2021-10-28  [id 1884]

(in reply to Report 1 on 2021-10-13)
Category:
remark

We thank the referee for their thorough reading and their positive feedback and we are very happy to hear that the referee enjoyed reading our manuscript. We revised the abstract, the introduction and title of our manuscript in order to further emphasize the strengths that the referee pointed out.

---

## Round 1 · Referee Report · Anonymous (Referee 2) · 2021-10-15

Strengths

1) Extremely thorough exposition of the topic. 2) Analysis is clear and correct. 3) Discuss role of imperfections and practical experimental realizations, which are generally ignored in this topic. 4) Present both detailed analytical results and comprehensive numerics.

Weaknesses

1) The paper presents many more details than are needed to understand the main results.
2) The writing style is a bit dry.

Report

The paper studies measurement-induced dynamics of bosonic systems with Gaussian evolution and measurements. As a result, similar to Clifford circuits in the case of qubits, there is a general purpose classical algorithm to solve for the dynamics. The authors present several models within this framework and study them analytically and numerically. Notably, they also consider the role of experimental imperfections, finding some level of robustness of their results to realistic levels of noise.

The paper is certainly worthy of publication. The analyses of imperfections is particularly interesting as many papers on this topic ignore this aspect of the problem.

Requested changes

1) Pg 1 - "integrable" should be "non-integrable" 2) The paper reads more like a report of an analysis, rather than conveying an important message that will motivate future work or solves an outstanding problem. It would be nice if the authors could attempt to convey the relevance of this work to a broader audience and for the future directions of the field.

  • validity: top
  • significance: good
  • originality: good
  • clarity: high
  • formatting: excellent
  • grammar: excellent

Author:  Yuri Minoguchi  on 2021-10-28  [id 1885]

(in reply to Report 2 on 2021-10-15)

We thank the referee for their careful reading of our manuscript and for pointing out its strengths and its weaknesses. We are happy to read the referee’s overall positive evaluation and we acknowledge their partial criticism on its accessibility for the readers. We took this criticism serious and have revised our manuscript, especially the abstract, the introduction and the title in order to make the manuscript more accessible. Our main messages are now conveyed more transparently and in a better structured way.

---

## Editorial Decision

resubmitted